# Intratumor graph neural network recovers hidden prognostic value of multi-biomarker spatial heterogeneity

Lida Qiu[1,2,9], Deyong Kang[3,9], Chuan Wang[4,9], Wenhui Guo[4], Fangmeng Fu[4], Qingxiang Wu [1], Gangqin Xi[1], Jiajia He[1], Liqin Zheng[1], Qingyuan Zhang[5], Xiaoxia Liao[6], Lianhuang Li [1✉], Jianxin Chen [1✉] & Haohua Tu[7,8✉]

Biomarkers are indispensable for precision medicine. However, focused single-biomarker development using human tissue has been complicated by sample spatial heterogeneity. To address this challenge, we tested a representation of primary tumor that synergistically integrated multiple in situ biomarkers of extracellular matrix from multiple sampling regions into an intratumor graph neural network. Surprisingly, the differential prognostic value of this computational model over its conventional non-graph counterpart approximated that of combined routine prognostic biomarkers (tumor size, nodal status, histologic grade, molecular subtype, etc.) for 995 breast cancer patients under a retrospective study. This large prognostic value, originated from implicit but interpretable regional interactions among the graphically integrated in situ biomarkers, would otherwise be lost if they were separately developed into single conventional (spatially homogenized) biomarkers. Our study demonstrates an alternative route to cancer prognosis by taping the regional interactions among existing biomarkers rather than developing novel biomarkers.

[1] Key Laboratory of OptoElectronic Science and Technology for Medicine of Ministry of Education, Fujian Provincial Key Laboratory of Photonics Technology, Fujian Normal University, Fuzhou 350007, China. [2] College of Physics and Electronic Information Engineering, Minjiang University, Fuzhou 350108, China. [3] Department of Pathology, Fujian Medical University Union Hospital, Fuzhou 350001, China. [4] Department of Breast Surgery, Fujian Medical University Union Hospital, Fuzhou 350001, China. [5] Department of Medical Oncology, Harbin Medical University Cancer Hospital, Harbin 150081, China. [6] National Center for Supercomputing Applications, University of Illinois at Urbana-Champaign, Urbana, IL 61801, USA. [7] Beckman Institute for Advanced Science and Technology, University of Illinois at Urbana-Champaign, Urbana, IL 61801, USA. [8] Department of Electrical and Computer Engineering, University of Illinois at Urbana-Champaign, Urbana, IL 61801, USA. [9] These authors contributed equally: Lida Qiu, Deyong Kang, Chuan Wang. ✉email: lhli@fjnu.edu.cn; chenjianxin@fjnu.edu.cn; htu@illinois.edu

Biomarkers improve patient care and impact therapeutics development[1]. The risk of a disease, e.g., recurrence of cancer, is routinely assessed by prognostic biomarkers[2]. The development of tissue prognostic biomarkers for breast cancer (exemplary of most diseases) has historically followed the one-biomarker-at-a-time approach, in which candidate biomarkers underwent representative multiregion sampling to suppress the spatial intratumor heterogeneity highlighted by multiregion sequencing and gene expression[3]. The multiregion sampling may take a virtual form such as multiple microscopic views of one histologic tissue section, or a physical form such as multiple microdissections from one tissue section, multiple tissue sections/blocks or tumor microarray cores from one primary tumor, and specific multisite sampling[4]. The former is frequently used in standard (H&E) histology and immunohistochemistry (IHC), whereas the latter in molecular profiling[5] (MP) such as multigene assays[6]. Regardless of the associated natures (morphological vs. molecular) and methods (H&E, IHC, and MP), single spatially homogenized biomarkers robust against the intratumor heterogeneity have been sequentially developed from underlying in situ biomarkers and included into various multivariate prognostic models for clinical use (Fig. 1a).

This homogenization of biomarkers is beneficial due to: (i) reliable pathology reports with less interobserver discordance; (ii) simple patient stratification for biomarker validation; and (iii) easy integration with other non-in-situ biomarkers into various prognostic models[7]. On the other hand, it is costly due to: (i) the discordances between primary tumors and distant metastases, e.g., discordance of estrogen receptor (ER) (or progesterone receptor PR) observed in 7–25% (or 16-49%) of breast cancer patients[8]; (ii) the subjectivity of disease-dependent criteria and thresholds, e.g., the histological grade based on averaged tubule formation but the highest mitotic count and degree of nuclear pleomorphism over sampled regions/views[9]; and (iii) small number of homogenized biomarkers in comparison to in situ biomarkers, which contributes to few reported biomarkers implemented into clinical practice[10]. A natural question arises whether the homogenization self-evidently leads to a high benefit-to-cost ratio, or whether it is self-evident to conduct breast cancer prognosis using the homogenized (rather than in situ) biomarkers (Fig. 1b).

In the simplest case of 1-to-1 correspondence between one homogenized biomarker and one in situ biomarker (e.g., regional percentage of ER + tumor nuclei vs. ER overall status, Fig. 1a), the benefits of the homogenization generally outweigh the corresponding costs because the candidate biomarker (ER) is not placed in the context of (does not spatially interact with) other biomarkers. However, ER overall status is rarely used as a standalone prognostic biomarker but routinely combined with other single homogenized biomarkers in a multivariate prognostic model. In this case of N-to-N correspondence between homogenized and in situ biomarkers, the balance of benefit-to-cost ratio may tip against the homogenization owing to the loss of information on implicit biomarker-biomarker interactions from multi-biomarker spatial heterogeneity (Fig. 1c). That is, multiple in situ biomarkers with co-registered multiregion sampling may derive differential prognostic value from these interactions over the multivariate non-graph prognostic model wherein the in situ biomarkers are developed into the corresponding homogenized biomarkers (Fig. 1c). This "hidden" prognostic value has not been demonstrated to date possibly due to: (i) technical difficulty to spatially co-register many in situ biomarkers among consecutive sections and different microscopic methods or scales (H&E and IHC); (ii) high cost of multiregion MP for a large cohort of patients; and (iii) lack of a computational representation for the biomarker-biomarker interactions.

We aim to demonstrate this value by representing the recently reported tumor-associated collagen signatures[11] (TACS1-8), i.e., multiple co-registered biomarkers from multiphoton microscopy (MPM), with a graph neural network[12] of primary breast tumor[13]. It should be noted that TACSs and other collagen-based biomarkers have been extensively studied[14] and explored for clinical translation[15]. The resulting prognostic model of intratumor graph neural network (IGNN) not only forms a high-performance kernel to include additional in situ biomarkers (from H&E, IHC, MP, etc.), but also provides *post hoc* interpretations of various implicit interactions among the included biomarkers with distinct biological/medical insights (Fig. 1a−c). Our distance-less IGNN model based on MPM may motivate the development of similar models based on other imaging and non-imaging tissue assessment technologies (Supplementary Table 1), as long as the biomarkers possess spatial heterogeneity.

## Results

**Construction of personized IGNN structure.** For a breast cancer patient characterized by one histologic formalin-fixed paraffin-embedded (FFPE) tissue block, the co-registration between H&E and MPM from two consecutive histologic tissue sections (4 µm) ensured the localization of several local MPM regions in global H&E-revealed tumor area and extraction of in situ TACS data[11]. One section was stained with H&E for whole slide imaging, in which a pathologist confirmed the presence of tumor cells and their borders. Dependent on the size of tumor area for sufficient sampling, several (4–20) ~2.8-mm-sized non-overlapping regions of interest (ROI) were located mainly at the tumor invasive front and then labeled (numbered) in the H&E images. The other (unstained) section was deparaffinized by alcohol and xylene to collect label-free dual-modal MPM of second harmonic generation (SHG) and two-photon excited (intrinsic) fluorescence (TPEF) images[16] for all labeled ROIs (Fig. 1d).

Based on the resulting personized MPM regions and heterogeneous regional distribution of TACS1-8 (8 in situ binary biomarkers), we constructed the IGNN to represent the MPM regions (nodes) with specific TACS1-8 distributions (node attributes present as 8-bit vectors) and their interactions (edges) with learnable parameters (Fig. 1d). This structure cast the prognostic prediction of many patients as a supervised Cox proportional hazards regression learning task by establishing a nonlinear mapping between TACS1-8 spatial heterogeneity and observed prognosis (Supplementary Fig. 1 and Supplementary Fig. 2). It consisted of multiple stacked nonlinear functional layers (Fig. 1d, Supplementary Fig. 3, and Supplementary Table 2), which were trained end-to-end by back propagation algorithm. First, graph convolution was used to represent intratumor graph nodes and infer their regional interactions. Two graph convolution layers followed neighborhood aggregation framework[17] in which the node attribute embeddings of graph structure were alternately updated by implementing message-passing mechanism[18]. Then, attention mechanism and optional gated recurrent units (GRU)[19] were introduced within graph convolution layers to capture more distinguishing features and reduce the gradient disappearance and over-smoothing problems during the model training phase. Later, the convoluted graph representation from all node attribute embeddings was further aggregated within a global pooling layer and abstracted by a fully connection layer. Finally, the graph representation was converted into IGNN score by a prognostic risk prediction layer to predict patient prognosis.

As a framework based on neural network, our IGNN might incorporate other quantifiable prognostic biomarkers as multi-modal input by adding the appropriate multilayer perceptron module[20] and feature fusion layer before the first full connection

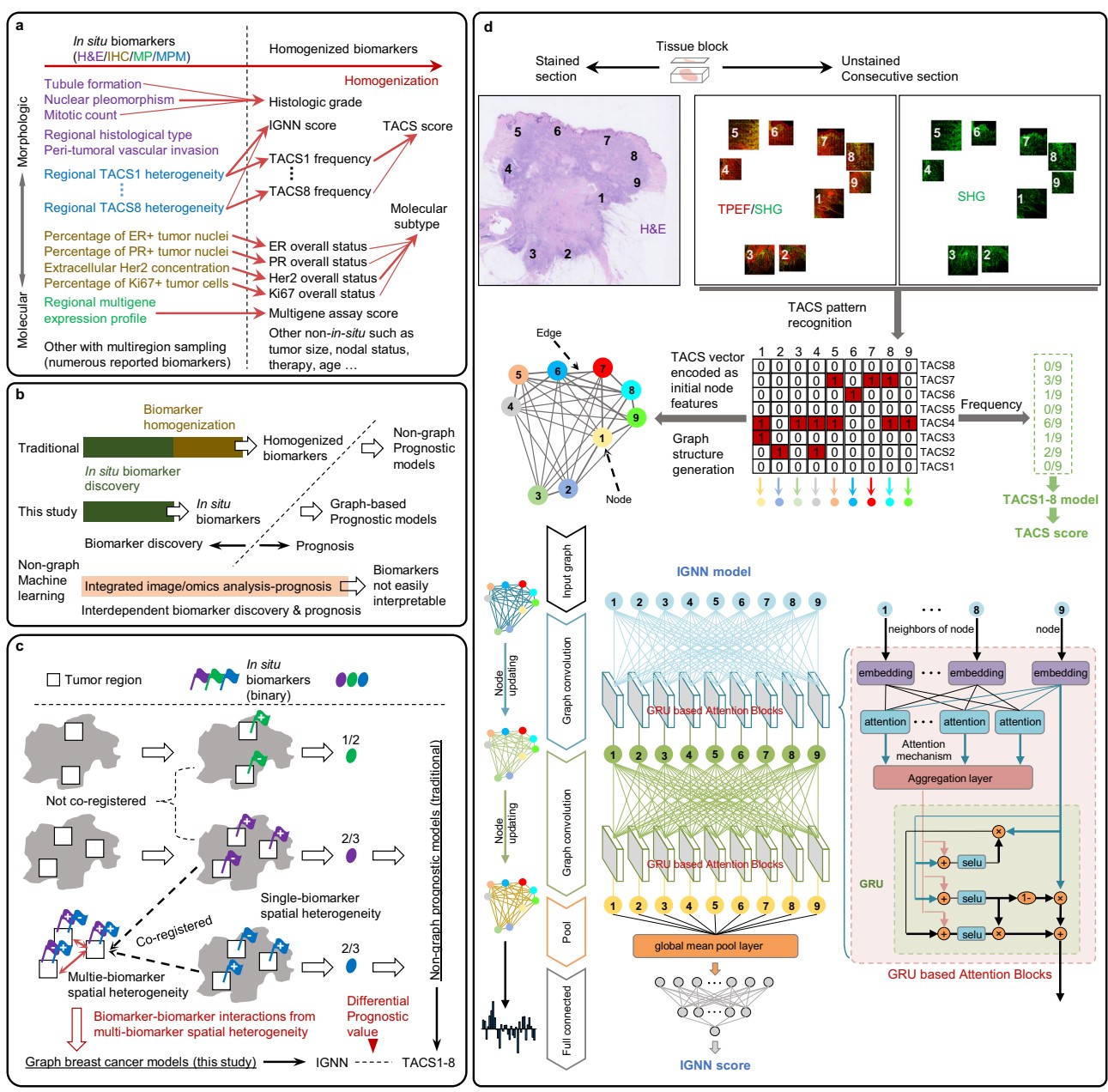

**Fig. 1 GNN representation of primary breast tumor and related prognostic models. a** Development of breast cancer prognostic biomarkers. **b** Main strategies of biomarker development for disease prognosis. **c** Co-registered multi-biomarker spatial heterogeneity (IGNN) in primary breast tumor with biomarker-biomarker interactions unavailable from corresponding single-marker heterogeneity (TACS1-8). **d** Personalized TACS1-8 reginal distributions in co-registered images (H&E, MPM of second harmonic generation SHG, and MPM of two-photon excited fluorescence TPEF) from one exemplary patient (9 regions/nodes, each of which encoded with an 8-bit vector) that result in one graph structure input for IGNN model and another non-graph input for TACS1-8 model. In the IGNN model that ends with an IGNN score, GRU based attention blocks execute node information propagation and aggregation along the graph structure and keep or remove node information related to prognostic prediction, whereas graph convolution and full connection extract prognostic graph representation and high-dimensional features sequentially. H&E hematoxylin and eosin, IHC immunohistochemistry, MP molecular profiling, ER estrogen receptor, PR progesterone receptor, HER2 human epidermal growth factor receptor 2, Ki67 specific nonhistone nuclear protein, MPM multiphoton microscopy, TPEF two-photon excited (intrinsic) fluorescence, SHG second harmonic generation, TACS (TACS1-8) tumor-associated collagen signatures, IGNN intratumor graph neural network, GRU gated recurrent units, SELU scaled exponential linear units.

layer. To evaluate whether combined IGNN score and traditional clinicopathological factors could synergistically improve the performance of cancer prognosis, we developed an extended model (IGNN-E) by incorporating clinical information (routine prognostic biomarkers) to the basic IGNN model (Supplementary Fig. 3 and Supplementary Table 3), just like how an extended model (Nomogram) of multivariate Cox proportional hazard regression was developed by incorporating this clinical

information to the basic TACS1-8 model[11,21]. The cost of biomarker homogenization was assessed by the differential prognostic value of the IGNN (or IGNN-E) model over the TACS1-8 (or Nomogram) model (Fig. 1c, d).

**Differential prognostic value from diverse patients.** We compared four prognostic models with different biomarkers (Supplementary Table 4). Using a training cohort of 731 patients, we

performed the pre-validation[22] based on 3-fold cross-validation for each of the four models (Supplementary Fig. 1). Personalized prediction scores for each instance held out during cross-validation were computed, and then all results from single cross-validation steps were combined as the model-specific validation results for the training cohort to assess their risk stratification capabilities (Supplementary Fig. 4 and Supplementary Table 5, see Methods). We subsequently retrained these models using the whole training cohort and finally validated the trained models using an independent validation cohort of 264 patients to further assess their prognostic value (Supplementary Fig. 1). For both cohorts in external validation, a correlation analysis was conducted between model scores and disease-free survival (DFS) through the Pearson correlation, which indicated stronger association of IGNN (or IGNN-E) score with DFS than TACS (or Nomogram) score (Supplementary Fig. 5a). The relative strength of various prognostic biomarkers to predict DFS, which was assessed by multivariate Cox proportional hazard regression analysis (Supplementary Table 6), indicated stronger dominance over other prognostic biomarkers by the IGNN score than the TACS score (Supplementary Fig. 6). Just like the TACS score, the IGNN score functioned as an independent prognostic factor along with tumor size, lymph node status, and molecular subtype (Supplementary Table 6).

We next calculated model-dependent survival curves in which patients in the training (or validation) cohort were stratified into low- and high-risk groups via the optimal cutoff point dictated by the two-sided log-rank test statistics in model training phase. In comparison to the TACS1-8 (or Nomogram) model, the IGNN (or IGNN-E) model produced more diverged survivals for low- and high-risk patients and more diverged distributions of predicted score for patients with DFS less versus more than 5 years (Supplementary Fig. 5b). The stronger prognostic strength of the IGNN (or IGNN-E) model over the TACS1-8 (or Nomogram) model was also revealed by larger values of hazard ratio (HR) (stratification ability), concordance index (C-index), integrated cumulative/dynamic AUC (iAUC), the associated area under receiver operating characteristic curves (AUC) (Fig. 2a), and specificity/sensitivity to predict 5-year DFS rate (Supplementary Table 7). The relatively large number of patients from rather disparate populations and institutions in southern (training cohort) and northern China (validation cohort) strengthened the statistical significance of the IGNN model free of new biomarkers.

All these results demonstrated the differential prognostic value of the IGNN over TACS1-8 model by simply accounting for the biomarker-biomarker interactions from multi-biomarker spatial heterogeneity. With no new biomarker beyond TACS1-8 (8 in situ biomarkers), our IGNN recovered this differential prognostic value from regional interactions among TACS1-8, which approximated the differential prognostic value from the combination of routine biomarkers (i.e., the Nomogram over TACS1-8 model) (Fig. 2a), particularly for patients with a small (<2 cm) tumor (Fig. 2b). These two differential prognostic values were largely additive from the TACS1-8 to IGNN-E model (Fig. 2a, b), indicating rather independent prognosis of these regional interactions from the routine prognostic biomarkers. Thus, the combined IGNN score and traditional clinicopathological factors synergistically empower cancer prognosis.

**Assessment on patient subgrouping and informed treatment.** To examine how the observed differential prognostic value benefited different subgroups of patients, we combined the patients in both cohorts ($n = 995$) and divided them into different subgroups according to various clinicopathological factors

(Supplementary Table 8). Interestingly, the observed differential prognostic value of patients with a small ≤2 cm tumor approximated that from the combination of routine biomarkers, whereas patients with a moderate 2–5 cm or large >5 cm tumor obtained a less significant result (Fig. 2b). The observed disparity in differential prognostic value was likely caused by the representative (nonrepresentative) multiregion sampling of a small (large) tumor using one section from a FFPE tissue block ($\sim 2 \times 2 \times 0.4 \ cm^3$). This disparity might be correlated with similar disparities observed from different histological grades, nodal statuses, and TNM-stages (Supplementary Fig. 7).

The single most surprising result was the large differential prognostic value from patients with a small tumor ($n = 445$). Consistently, the dominance of the IGNN score (in contrast to the TACS score) over other prognostic biomarkers was more pronounced in this subgroup of patients than the whole group (Fig. 2c vs. Supplementary Fig. 6). This implied that the recovery of previously neglected regional biomarker-biomarker interactions among existing biomarkers could empower cancer prognosis more than the addition of new prognostic biomarkers. Thus, for this subgroup of patients, in situ biomarkers should be chosen over homogenized biomarkers for more accurate cancer prognosis.

To evaluate the benefit of the IGNN model beyond risk assessment, we investigated the corresponding utility to inform the postoperative adjuvant therapy of all 995 patients according to the consensus treatment guideline[23]. Both TACS score and IGNN score reclassified a significant portion of low-risk (high-risk) patients according to the guideline into high-risk (low-risk) patients, and thus could minimize undertreatment (overtreatment) (Fig. 3a). Based on the retrospectively determined 5-year DFS rate, in contrast to the TACS score that might spare 125/112 patients from overtreatment/undertreatment according to the guideline, the IGNN score would spare 138/106 patients from this undertreatment/overtreatment (Fig. 3b). As to the 445-patient subgroup, in contrast to the TACS score that might overtreat 94 patients, the IGNN score would overtreat 52 patients and thus spare 42 patients from this overtreatment (Fig. 3c, d).

To sum up, the shift from the TACS to IGNN model not only empowered cancer prognosis but also could improve subsequent informed treatment, particularly for patients with a small tumor (representative multiregion sampling).

**Post hoc interpretations of biomarker-biomarker interactions.** How the IGNN model learns the implicitly prognostic interactions among TACS1-8 is valuable to understand the TACS1-8 themselves beyond morphology. The message passing of the graph convolutional layers allows nodes to progressively aggregate the most relevant messages from their neighborhoods via the attention mechanism. In the training process, observable and understandable model responses are important to investigate how the node states evolve in individual patients. Without loss of generality, we randomly interrupt a training process of the IGNN model after sufficient training without any prior knowledge (see Methods), then obtain the corresponding feature vectors from the input and graph convolutional layers of the trained IGNN (Fig. 1d) and compute the similarity between regional pairs through the heat map of Pearson correlation coefficient matrix (Supplementary Fig. 8). The initial node state is encoded according to regional presence of TACS(s). The correlation of the regional pair with the initial state reflects the extent to which they contain the same TACS(s). However, as nodes and their neighbors interact through the graph convolutional layers, the node state vectors are constantly updated, resulting in the correlation changes among the nodes.

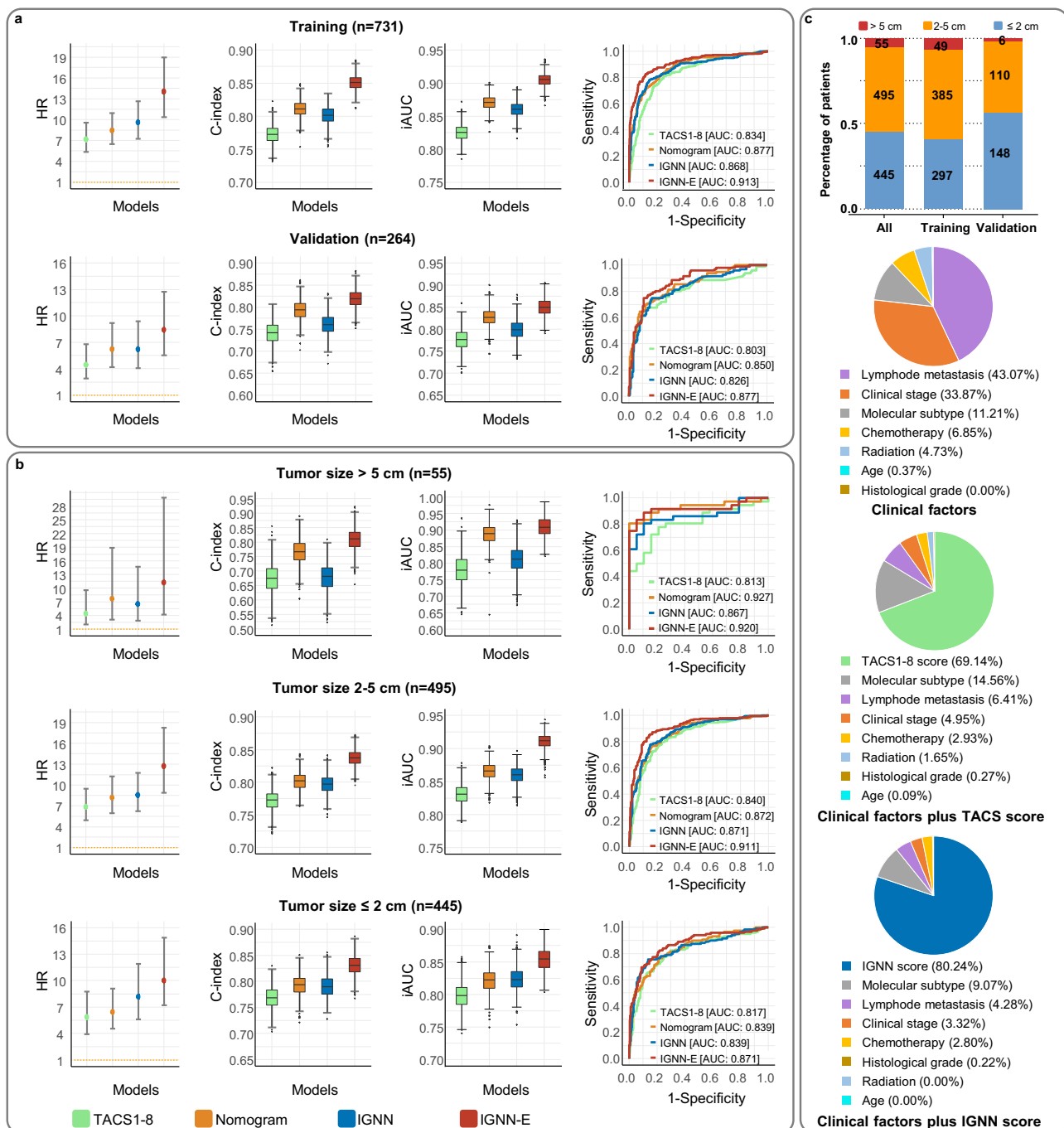

**Fig. 2 Performance of four prognostic models in external validation. a, b** HR from multivariate Cox proportional hazards regression analysis (in error bars, circles and upper/lower boundaries indicate mean value and 95% CIs, respectively. A two-sided log-rank test was performed to determine significance), distributions of C-index and iAUC from time-dependent AUC analysis (in boxplots, middle line represents the median value, the upper and lower boundaries of boxes indicate 25th and 75th percentile, the whiskers reflect 1.5 times of interquartile range, the upper and lower tails indicate the maxima and minima, and single points indicate the outliers. A two-sided unpaired t-test was performed to determine significance), and receiver operating characteristic (ROC) curves with AUC, for different groups of patients. **c** Tumor size distribution for patient cohorts (upper panel) and relative contributions of prognostic biomarkers in predicting DFS of 445 patients with a < 2 cm tumor from multivariate Cox proportional hazard regression analysis (lower panels). Chi-squared test was performed to determine significance. HR hazard ratio, C-index concordance index, AUC associated area under receiver operating characteristic curves, iAUC integrated cumulative/dynamic AUC, TACS (TACS1-8) tumor-associated collagen signatures, Nomogram, extended model of multivariate Cox proportional hazard regression, IGNN intratumor graph neural network, IGNN-E extended IGNN model with clinical information. Source data are provided as a Source Data file.

We apply the above method to the training cohort and find that node (ROI) state feature output from graph convolutional layer 2 exhibits distinct pattern and order according to regional interactions among TACSs. The heat map clearly divides thousands of MPM regions from 731 patients into three clusters. The regions with TACS5 or TACS6 accounts for 79% in Cluster 1, the regions with TACS4 accounts for 90% in Cluster 2, and the regions with TACS1 accounts for 92% in Cluster 3 (Fig. 4a).

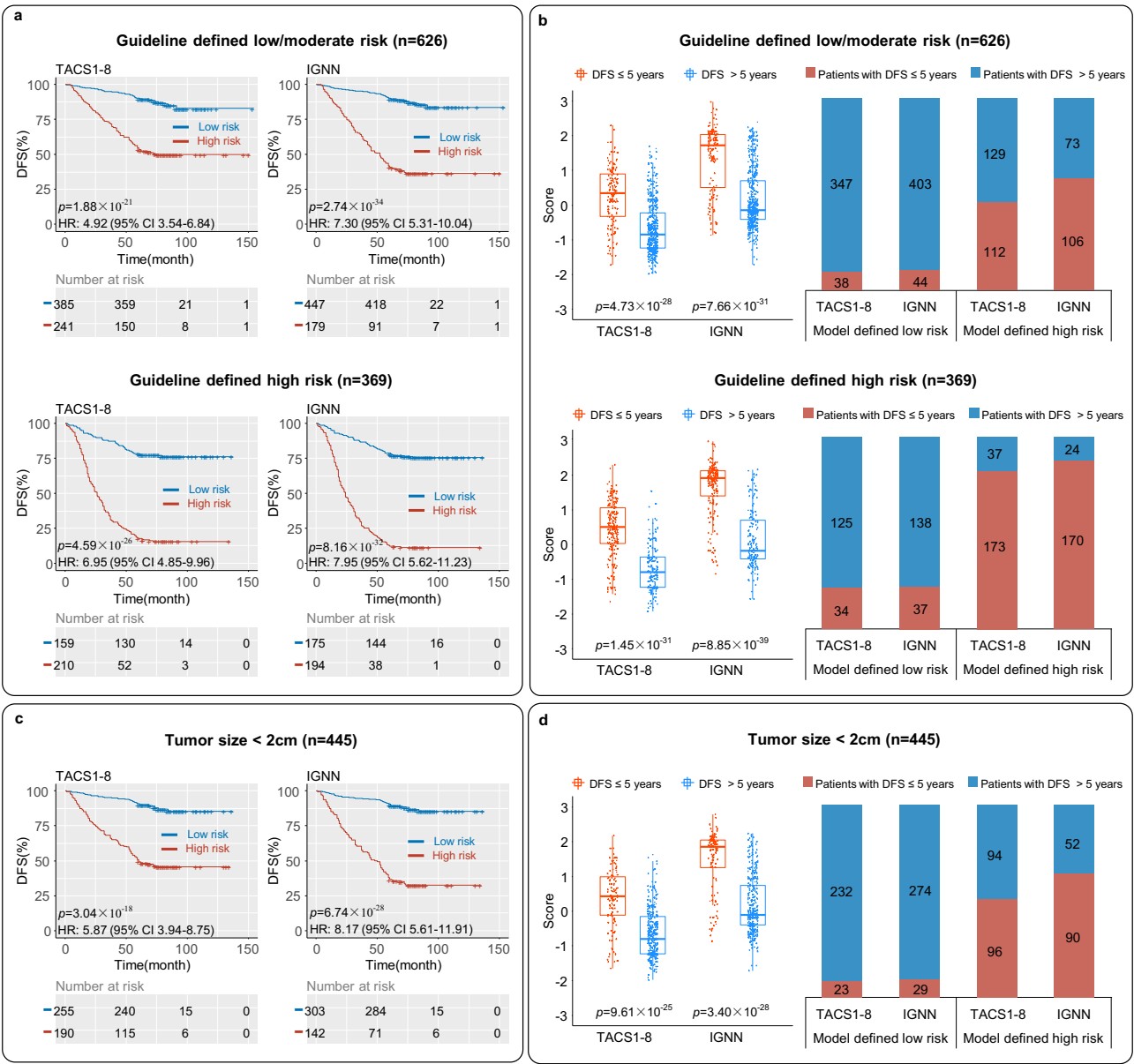

**Fig. 3 Comparison between IGNN and TACS1-8 models in external validation. a** Kaplan-Meier survival analysis of patients with the treatment guideline-derived low/moderate- and high-risk, re-stratified into high (red line) and low risk group (blue line) based on two prognostic models. A two-sided log-rank test was performed to determine significance, showing HR values with confidence intervals and exact *p* values. **b** Corresponding boxplots (left panels) showing TACS and IGNN scores for patients with DFS less and more than 5 years (in boxplots, middle line represents the median value, the upper and lower boundaries of boxes indicate 25th and 75th percentile, the whiskers reflect 1.5 times of interquartile range, the upper and lower tails indicate the maxima and minima, and single points indicate the distribution of values. A two-sided unpaired t-test was used to compare risk subgroups and determine significance from exact *p* values), and percentage histogram (right panels) showing survival distributions for model-based risk subgroups indicative of undertreatment and overtreatment. **c**, **d** Similar information obtained from 445 patients with a small (<2 cm) tumor. HR hazard ratio, DFS disease-free survival, TACS (TACS1-8) tumor-associated collagen signatures, IGNN intratumor graph neural network. Source data are provided as a Source Data file.

Analysis of similarities (ANOSIM) of the output feature vectors from the graph convolutional layer 2 for the nodes (ROIs) shows significant differences between the three clusters (Fig. 4a). Similar results are obtained from the validation cohort (Fig. 4b) and patients with a small tumor (Fig. 4c), indicating that the IGNN model captures prognostic value of TACS1-8 interactions by learning from a specific task.

A number of *post hoc* interpretations of TACS1-8 emerge in consistency with their morphologies (Fig. 4d): (i) high positive correlation between regions with TACS5 and TACS6 indicates their converging function and possibly synergistic interaction in

tumor cell invasion; (ii) low correlation between regions with TACS4 and TACS5,6 implies that their effects on tumor cell invasion are relatively independent or different (TACS4 appears to block tumor cell invasion); (iii) low or negative correlations between regions with TACS1 and regions with TACS4,5,6 confirm that TACS1 is characteristic more of tumorigenesis than tumor cell invasion; and (iv) no clusters with high correlation between intra-cluster regions contain a high percentage of regions with TACS2, TACS3, TACS7, and TACS8, which may be due to the relative rarity of TACS2,3,7,8 in the samples. Additionally, the proportions of TACS2,3,7,8 are unevenly distributed across

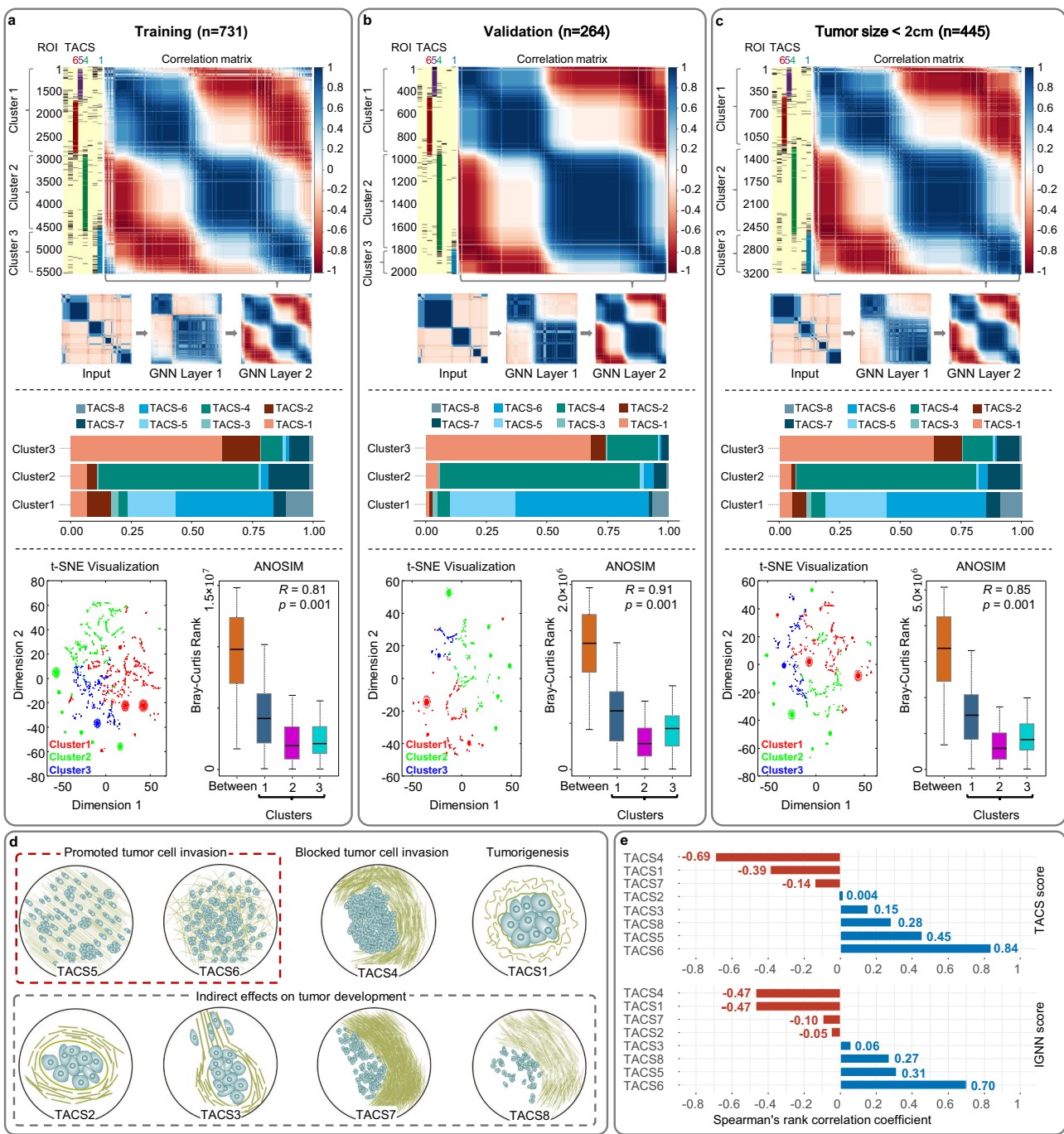

**Fig. 4 Implicit interactions among TACSs interpreted by IGNN learning. a**, **b**, **c** Automatic separation of patients into three clusters by the similarity of node state output from graph convolution layer 2 using Pearson correlation matrices of IGNN nodes (top), percentages of TACSs in the clusters (middle), learned node features projected to two dimensions for visualizing the clusters via t-distributed stochastic neighbor embedding (t-SNE technology) (bottom left), and ANOSIM that reveals the dissimilarities between the samples from different clusters (bottom right). In ANOSIM, the boxplots marked by "Between", "1", "2" and "3" show the Bray-Curtis rank distribution for ROI sample pairs between two clusters under comparison and within clusters 1-3, respectively. In boxplots, middle line represents the median value, the upper and lower boundaries of boxes indicate 25th and 75th percentile, the whiskers reflect 1.5 times of interquartile range, and the upper and lower tails indicate the maxima and minima. Wilcoxon rank sum test was performed to compare the significant difference between clusters without any adjustments for multiple comparisons, with statistic $R$ ranges between the values −1.0 to 1.0, wherein a value close to 1.0 suggests dissimilarity between clusters while a value <0 suggests greater dissimilarities within clusters than between clusters (observed $R$ value over the null distribution was used to assess $R$ statistic from exact $p$ values, and the significance was defined as $p < 0.05$). **d** *Post hoc* interpretations of the nature of TACS1-8 and their regional/spatial interactions. **e** Correlation analysis between individual TACSs and TACS score and IGNN score. ROI regions of interest, TACS (TACS1-8) tumor-associated collagen signatures, GNN graph neural network, IGNN intratumor graph neural network, t-SNE t-distributed stochastic neighbor embedding, ANOSIM Analysis of similarities. Source data are provided as a Source Data file.

different clusters (Fig. 4a–c, middle), and ANOSIM shows that ROIs with TACS3 or TACS8 correlate more with Cluster 1 than other clusters while ROIs with TACS2 (TACS7) correlate more with Cluster 3 (Cluster 2) (Supplementary Fig. 9). Considering collagen morphologies, we hypothesize that TACS2,3,7,8 affect survival by interacting with other TACSs. For example, sparsely distributed collagen fibers at the tumor invasion front (TACS8) is conducive to TACS5,6 that enables tumor cell migration, while densely distributed collagen fibers at the tumor invasion front (TACS7) is conducive to TACS4 that restricts tumor cell migration. It should be noted that these insightful interpretations are not available from the TACS1-8 model with homogenized TACSs[11].

Another *post hoc* interpretation arises from the correlation analysis between individual TACSs and TACS/IGNN score, which shows a generally lower correlation of individual TACSs with IGNN score in comparison to the TACS score (Fig. 4e). Because the IGNN model accounts for the interactions among TACSs, the prognostic importance (absolute values of Spearman's rank correlation coefficients) of TACSs becomes less in the IGNN model vs. TACS1-8 model (Fig. 4e).

## Discussion

The results demonstrated in this study promote a shift of disease prognosis from conventional (spatially homogenized) biomarkers to in situ biomarkers that reflect intratumor heterogeneity (Fig. 1a, b). Also, for existing homogenized biomarkers (e.g., ER in Fig. 1a), the corresponding in situ biomarkers may be added to TACS1-8 to further improve the IGNN-E model (Fig. 2a). The required co-registration of in situ biomarkers among MPM, IHC, and H&E can be ensured by collecting and examining consecutive FFPE tissue sections. Moreover, the individual biomarkers of IGNN may be expanded from binary biomarkers (TACS1-8) to those with multiple or continuous values (e.g., regional percentage of ER+ tumor nuclei, Fig. 1a), in order to retain full prognostic information. Finally, further optimization of region size (2.8 mm in this study) for more informative spatial biomarker distribution (Supplementary Fig. 10) and optimization of model depth for a larger patient size may increase the performance of prognosis. Thus, cost-effective precision medicine may be obtained by taping current underappreciated biomarkers rather than developing novel biomarkers, because the hidden prognostic value of regional biomarker-biomarker interactions can be recovered by an IGNN model.

This study exploits intratumor heterogeneity to improve biomarker utility, defying the conventional wisdom that intratumor heterogeneity is dismal for biomarker development[24]. A few other studies attempted to derive prognostic values from the spatial heterogeneity of cellular[25] or genetic biomarkers[26,27]. However, whether biomarker intratumor heterogeneity can serve as a practical prognostic biomarker remains controversial[28,29]. Here we avoid this controversy by deriving prognostic values from regional biomarker distributions (TACS1-8 model) and biomarker-biomarker interactions (IGNN model) originated from the intratumor heterogeneity, rather than from the diversity of regionally distributed TACS1-8 corresponding to the intratumor heterogeneity itself.

As an early attempt to exploit the benefits of intratumor heterogeneity, we have chosen a size of ~2.8 mm for ROIs, each of which produces ~1.3 distinct features of TACS1-8 (Supplementary Table 9). For this size, representative sampling of the tumor invasion front of a typical histological section requires 4-20 ROIs, depending on the dimension of the tumor area. The number of ROIs seem adequate because random removal of 20% of ROIs for

individual patients results in rather small inconsistency in prognosis (Supplementary Table 10). Thus, it is unlikely that our particular choice of ROIs (number and position) introduces an arbitrary level of intratumor heterogeneity. However, our combination of ROI size, sampling site (invasion front), and number of ROIs may not be optimal to attain the full potential of the IGNN model. Future efforts to balance the trade-off between ROI size (content of TCASs in one region) and ROI number (content of TACS-TACS interaction among regions), and the trade-off of overall TACS content versus cost, may yield better overall performance in a clinical setting. The extension of TACS content from tumor invasion front to tumor center may further improve the performance.

Our multi-biomarker IGNN enables unique integrative research on tumor microenvironment from the perspective of cancer risk. The corresponding model based on MPM-revealed extracellular matrix remodeling[30] achieves a high performance (Figs. 2 and 3) among tumor microenvironment-centered prognostic models based on MP-revealed stromal gene expression[31], H&E-revealed stromal architectures[32], H&E-revealed tumor-infiltrating lymphocytes[33] or stromal cells[34], and IHC-revealed specific immune cells[35]. This high performance may arise from the general role of extracellular matrix in modulating the hallmarks of cancer[36]. High-performance cancer prognosis has not been attained in previous studies[14,37] on TACS1-3 and aligned collagen possibly due to their emphasis on developing single and/or homogenized biomarkers. To further improve the IGNN model, it is reasonable to complement TACS1-8 with the above tumor microenvironment components by co-registering the corresponding biomarkers (Fig. 1c). A more complete picture of tumor microenvironment should further include tumor cell components (e.g., histological or molecular type discussed above). The crosstalk among various tumor cell and microenvironment components may be interpreted *post hoc* from the regional biomarker-biomarker interactions (Fig. 4). It will be interesting to compare the independent generation of prognostic values under the conditions of single- versus multiple-biomarker spatial heterogeneity (Fig. 1c).

The graph-based prognostic model improves the performance of its conventional non-graph counterpart, and at the same time, retains or enhances the interpretability of the underlying biomarkers. In contrast to various prognostic morphological biomarkers derived from graph-based[38] and non-graph machine/deep learning in computational histopathology[39–42], our in situ biomarkers (regional heterogeneity of TACS1-8) were identified in a discovery process[11] independent of subsequently developed IGNN prognostic model (Fig. 1b), so that they are more amendable for human interpretations (Fig. 4). Also, the contextual information connecting sampling regions are conveniently represented by the edges connecting the nodes (Fig. 1d), relieving the somewhat challenging task in typical deep learning to model the spatial correlations between neighboring patches (by combining low- and high-resolution inputs)[43]. Our IGNN significantly departs from prior GNN research that centers graph nodes on cells[44], which assumes that tumor cell-cell distances and interactions have the highest prognostic value[13]. Strikingly, the "functional" interactions among TACSs recover a differential prognostic value larger than that of combined routine prognostic biomarkers for small tumors (Fig. 2b), without accounting for internode distances in the graph edges (see Methods). It is reasonable to assume that future inclusion of these distances may further improve the performance of cancer prognosis, particularly that associated with large tumors after more representative multiregion sampling.

One perceived weakness of our approach to manually recognize TACS1-8 may be overcome by an automatic recognition procedure after weakly supervised deep learning[45]. Also, our IGNN model is restricted to adjuvant therapy patients, and is not applicable to neoadjuvant therapy patients whose tumors and TACS features have been perturbed by the neoadjuvant therapy[46]. Moreover, the insufficient sampling of a tumor with specific selection of ROIs may introduce prognosis uncertainty at individual level (but not necessarily population level) due to uncertain intratumoral heterogeneity such as stroma versus cellularity ratio. On the other hand, our IGNN highlights an overlooked advantage of biospecimen sampling using multiple laser-capture microdissections and tissue microarray cores from one sample over homogenized tissue sampling[47] and liquid biopsy. Although this study is limited to a specific disease (breast cancer) and prognostic prediction (predicting DFS), similar methodology can be adopted for other diseases and prognostic predictions, e.g., predicting treatment response to a therapy/drug.

## Methods

**Patients, samples, image acquisition, and TACS recognition**. This research used anonymous data for retrospective study and was conducted under a protocol approved by the Institutional Review Boards (IRB) of Fujian Medical University Union Hospital and Harbin Medical University Cancer Hospital. Our study included a training cohort of 731 patients from Fujian Medical University (FMU) Union Hospital and a validation cohort of 264 patients from Harbin Medical University (HMU) Cancer Hospital. The demographic and clinicopathologic characteristics of patients, sample preparation, MPM image acquisition, and TACS recognition have been reported previously[11]. Only a small portion (~5%) of patients has multifocal tumors, and in this case, the sample preparation was performed on one of the foci with the largest size. Raw data including TACS coding observed from MPM imaging, clinical factors and follow-up information of patients is available with this paper.

**Graph dataset**. As typical graph machine learning models, IGNN and IGNNE were implemented with specific irregular graph structures as input. We developed a toolset based on the PyTorch-Geometric library to build special graph structures from raw data of patients and generate specialized graph dataset (TACS_G) containing node-level attributes and connectivity matrix from a batch of graph structures, and the toolset and pre-established TACS_G dataset for this study are available within the source code.

**Personalized IGNN structure**. For each given tissue section (patient), TACS distribution is encoded in a graph data structure defined as G = (V, E), which consists of an $N$-node set V and an edge set E. Each ROI in MPM image in the set $\text{ROI}_{set} = \{\text{ROI}_1, \text{ROI}_2, \ldots, \text{ROI}_N\}$ is regarded as a node object in V. An 8-bit binary coding vector $\mathbf{v}_i \in \mathbb{R}^{1 \times 8}$ forms the initial feature vector of the $\text{ROI}_i$-associated node $i \in V$ to describe the TACS1-8 distribution in $\text{ROI}_i$ (e.g., $\text{ROI}_i$ including TACS4 and TACS7 resulted in $\mathbf{v}_i = 01001000$), and $\mathbf{F} = [\mathbf{v}_1^T, \cdots, \mathbf{v}_N^T]^T \in \mathbb{R}^{N \times 8}$ is defined as the initial node feature matrix of G. The edge $e_{ij} \in E$ between node $i$ and $j$ is then constructed according to the $k$-nearest neighbor algorithm

$$e_{ij} = 1, \text{if } j \in \text{KNN}(i) \text{ and } \mathbf{v}_i \cdot \mathbf{v}_j^T \geq 0 \tag{1}$$

where the inner product $\mathbf{v}_i \cdot \mathbf{v}_j^T[0, \cdots, 8]$ indicates the number of TACS types presented in both $\text{ROI}_i$ and $\text{ROI}_j$ and $\text{KNN}(i)$ indicates $k$-nearest neighbors of node $i$. Finally, personized TACS information is converted into an IGNN structure.

**Architectures of IGNN and IGNN-E models**. The IGNN model takes the IGNN graph structure data as input and outputs a score for prognostic prediction (IGNN score). Given a set of G = {$G_1, G_2, \ldots, G_N$} with corresponding survival status labels Y = {$y_1, y_2, \ldots, y_N$}, the IGNN model constructs a multilayer nonlinear mapping to extract significant representation from graph structure data for prognostic prediction. It contains three components: (a) graph convolution network module; (b) fully connected network modules; and (c) prognostic regression layer.

The core part of graph convolution network module contains multiple stacked graph convolution layers, designed as a "message-passing" architecture that drives nodes to aggregate information with their neighbors along edges and perceives their interactions. The graph convolution is separated into two steps: node information propagation and node information aggregation. In the first step, node features are propagated as the information of node state along the edge between neighboring nodes. Attention mechanism is incorporated into the propagation step, which follows a self-attention strategy to aggregate the neighboring node states of node $i$ by attending over them with different weights. The propagation

scheme conducted in the attention layer is formulized as

$$\mathbf{X}_i^{(t)} = \sum_{j \in N(i)} a_{ij} \mathbf{H}_j^{(t)} \tag{2}$$

where the set $N(i) = \{j \in V | e_{ij} = 1\}$ represents the set of 1-hop neighbors of node $i \in V$ in the graph, $\because$ represents the calculated weighted aggregation result of state information propagated from the neighbors of node $i$, $\mathbf{H}_j^{(t)} \in \mathbb{R}^{1 \times c^{(t)}}$ represents the $c^{(t)}$-dimensional feature vector of state information for node $j \in V$ in the $t$-th graph convolutional layer, $a_{ij}$ is the attention-based weight coefficient of state information propagated by node $j$ to node $i$ according to the attention mechanism of

$$a_{ij} = \frac{\exp(\beta^{(t)} \langle \mathbf{H}_i^{(t)} \mathbf{W}_H^{(t)}, \mathbf{H}_j^{(t)} \mathbf{W}_H^{(t)} \rangle)}{\sum_{k \in N(i)} \exp(\beta^{(t)} \langle \mathbf{H}_i^{(t)} \mathbf{W}_H^{(t)}, \mathbf{H}_k^{(t)} \mathbf{W}_H^{(t)} \rangle)} \tag{3}$$

where $\beta_i^{(t)} \in \mathbb{R}$ is learnable normalized weight adjustment parameter, $\mathbf{W}_H^{(t)} \in \mathbb{R}^{c^{(t)} \times c^{(t)}}$ is the learnable weight matrix shared among all the nodes in $t$-th graph convolutional layer, $\langle \cdot \rangle$ denotes the inner product operator on two vectors. In the second step, each node in the graph updates the current state $\mathbf{H}_i^{(t+1)}$ by aggregating past state information from itself ($\mathbf{H}_i^{(t)}$) and its neighbors ($\mathbf{X}_i^{(t)}$). The attention weight $a_{ij}$ are assigned to different nodes and edges according to the underlying dependencies to direct the network to the most prognostic parts of the TACS-based graph structure.

In addition, an optional aggregator based on Gated Recurrent Units (GRU) is designed to alleviate the potential over-smoothing issue during the information aggregation process as the network depth increases

$$\mathbf{H}_i^{(t+1)} = \begin{cases} \text{GRU}(\mathbf{H}_i^{(t)}, \mathbf{X}_i^{(t)}), \text{GRU is available} \\ \mathbf{H}_i^{(t)} + \mathbf{X}_i^{(t)}, \text{GRU is unavailable} \end{cases} \tag{4}$$

and the basic recursive formula of GRU aggregator is

$$\begin{aligned} \mathbf{H}_i^{(t+1)} &= \mathbf{Z}^{(t)} \odot \widetilde{\mathbf{H}}_i^{(t+1)} + (1 - \mathbf{Z}^{(t)}) \odot \mathbf{H}_i^{(t)} \\ \widetilde{\mathbf{H}}_i^{(t+1)} &= \text{Tanh}\left(\mathbf{X}_i^{(t)} \mathbf{W}_{xh} + (\mathbf{R}^{(t)} \odot \mathbf{H}_i^{(t)}) \mathbf{W}_{hh} + \mathbf{b}_H\right) \\ \mathbf{Z}^{(t)} &= \text{Selu}\left(\mathbf{X}_i^{(t)} \mathbf{W}_{xz} + \mathbf{H}_i^{(t)} \mathbf{W}_{hz} + \mathbf{b}_Z\right) \\ \mathbf{R}^{(t)} &= \text{Selu}\left(\mathbf{X}_i^{(t)} \mathbf{W}_{xr} + \mathbf{H}_i^{(t)} \mathbf{W}_{hz} + \mathbf{b}_R\right) \end{aligned} \tag{5}$$

where $\odot$ denotes elementwise multiplication, Tanh $(\cdot)$ is hyperbolic tangent nonlinear activation function, Selu $(\cdot)$ is the nonlinear activation function termed as scaled exponential linear units, $\mathbf{W}_{xh}, \mathbf{W}_{xz}, \mathbf{W}_{xr}, \mathbf{W}_{hh}, \mathbf{W}_{hz}, \mathbf{W}_{hr} \in \mathbb{R}^{c^{(t)} \times c^{(t)}}$ and $\mathbf{b}_H, \mathbf{b}_Z, \mathbf{b}_R \in \mathbb{R}^{1 c^{(t)}}$ indicate learnable weight matrixes and bias parameters, respectively. The reset gate denoted as $\mathbf{R}^{(t)} \in \mathbb{R}^{1 c^{(t)}}$ is used to control how the previous node state is merged into the current candidate node state, and update gate denoted as $\mathbf{Z}^{(t)} \in \mathbb{R}^{1 c^{(t)}}$ is used to control how node states are updated by combining past states with current candidate state. Node information propagation and aggregation is performed hierarchically along the layers. The $t$-th graph convolutional layer takes the node state information feature matrix $\mathbf{H}^{(t)} = [(\mathbf{H}_1^{(t)})^T, (\mathbf{H}_2^{(t)})^T, \cdots, (\mathbf{H}_N^{(t)})^T]^T \in \mathbb{R}^{N \times c^{(t)}}$ as input, and outputs a new feature matrix $\mathbf{H}^{(t+1)} = [(\mathbf{H}_1^{(t+1)})^T, (\mathbf{H}_2^{(t+1)})^T, \cdots, (\mathbf{H}_N^{(t+1)})^T]^T \in \mathbb{R}^{N \times c^{(t)}}$ updated via node information propagation and aggregation operations. Specially, in the non-linearity mapping process for converting the initial node state vectors $\mathbf{F} = [\mathbf{v}_1^T, \cdots, \mathbf{v}_N^T]^T \in \mathbb{R}^{N \times 8}$ into higher-level node feature embedding $\mathbf{H}^{(1)} = \text{Selu}(\mathbf{F}\mathbf{W}_f)$, the feature vectors of $\mathbf{F}$ are weighted by a learnable weight matrix $\mathbf{W}_f \in \mathbb{R}^{8 \times c^{(t)}}$ shared among all the nodes to highlight their relative importance in $\mathbf{H}^{(1)}$. The node states are propagated and aggregated across the multiple graph convolutional layers, which lead to the high-level node state features. A global pooling layer follows the last graph convolution layer to generate graph-level output for a single graph G by averaging all node state features in G across the node dimension

$$\mathbf{f}^{(l)} = \sum_{i=1}^{N} \mathbf{H}_i^{(t)} \tag{6}$$

where $\mathbf{f}^{(l)} \in \mathbb{R}^{1 \times c^{(t)}}$ is computed as the comprehensive representation of the graph structure, which extracts multi-scale localized patterns of substructure overall G. The Global pooling layer is followed by the fully connected network module consisting of multiple fully connected layers. Fully connected network module weights the portions of feature $\mathbf{h}^{(t)}$ with different importance levels via a learnable weight matrix $\mathbf{W}_h^{(t)}$ and recombines them into more discriminative higher-level feature $\mathbf{h}^{(t+1)}$ by the non-linearity mapping of

$$\mathbf{h}^{(t+1)} = \text{Selu}\left(\mathbf{h}^{(t)} \mathbf{W}_h^{(t)} + \mathbf{b}_h^{(t)}\right) \tag{7}$$

where $\mathbf{h}^{(t)} = \mathbb{R}^{1 \times h^{(t)}}$ and $\mathbf{h}^{(t+1)} = \mathbb{R}^{1 \times h^{(t+1)}}$ are the input and output feature vectors of the $t$-th full connected layer, respectively, $\mathbf{W}_h^{(t)} \in \mathbb{R}^{h^{(t)} \times h^{(t+1)}}$ and $\mathbf{b}_h^{(t)} \in \mathbb{R}^{1 \times h^{(t+1)}}$ are the learnable connected weight matrix and bias parameter, respectively. In particular, in the original architecture of IGNN, $\mathbf{h}^{(1)} = \mathbf{f}^{(l)}$ was used as the input of the first full connected layer, while in the IGNN-E model, the feature vectors made

up of 8 clinicopathological factors are cascaded with $\mathbf{f}^{(l)}$ as the input of the first fully connected layer

$$\mathbf{h}^{(l)} = \begin{cases} \mathbf{f}^{(l)}, & \text{IGNN} \\ \mathbf{f}^{(l)} \cup \text{Selu}\,(\mathbf{r}\mathbf{W}_r), & \text{IGNN} - \text{E} \end{cases} \qquad (8)$$

where $\bigcup$ concatenates two feature vectors along an assigned axis, $\mathbf{W}_r \in \mathbb{R}^{6 \times r}$ is the learnable connected weight matrix, $\mathbf{r}$ represents additional clinicopathological vectors, $\mathbf{h}^{(l)} = \mathbb{R}^{1 \times h^{(l)}}$ denotes the output high-level feature vector of the fully connected layer. Finally, these features are fed into the prognostic risk prediction layer along with survival data to produce the prediction score

$$p = \text{Selu}\,(\mathbf{h}^{(l)}\mathbf{W}_{\text{risk}}) \qquad (9)$$

$\mathbf{W}_{\text{cox}} \in \mathbb{R}^{h^{(l)} \times 1}$ is learnable weight vector.

The IGNN model is currently a small-scale architecture consisting of only 2 graph convolution layers and 2 fully connected layers, where the learnable weight matrixes $\mathbf{W}_H^{(t)}, \mathbf{W}_{xh}, \mathbf{W}_{xz}, \mathbf{W}_{xr}, \mathbf{W}_{hh}, \mathbf{W}_{hz}, \mathbf{W}_{hr}$ in graph convolutional layers have fixed size (8,8) and each fully connected layer consists of 32 neural units. As to the IGNN-E model, the weight matrix $\mathbf{W}_r$ applied to the additional feature vector $\mathbf{r}$ has a size of (8,16). The number of parameters for the IGNN (IGNN-E) model is ~2300 (~2500). The more learnable parameters of the model, the more training data are required to avoid the over-fitting in machine learning. It is thus reasonable to design a small-scale architecture for a relatively small patient number (995 in total). However, with increasing number of patients for training, the original models can be extended to more complex depth models by stacking more graph convolutional layers and fully connected layers.

To evaluate prognostic risk, the learning object of IGNN (IGNN-E) was to minimize the negative log partial likelihood of Cox proportional hazards regression loss as follow

$$\mathcal{L}(\boldsymbol{\theta}, \mathbf{b}) = -\sum_{i:y_i=1} (p_i - \log(\sum_{j:D_j \geq D_i} \exp(p_j))) \qquad (10)$$

where survival status label $y_i = 1$ indicates the occurrence of disease-related events (recurrence or death) for patient $i$, $D_i$ denotes the DFS of patient $i$, $\boldsymbol{\theta}$ and $\mathbf{b}$ are the set of learnable weights and biases, respectively.

**Training and validation**. In the pre-validation process, 3-fold cross-validation was adopted to train and evaluate the prognostic models within the training cohort (FMU dataset). The training cohort was first divided into 3 folds ($n_{\text{fold1}} = 243$, $n_{\text{fold2}} = 244$, $n_{\text{fold3}} = 244$). Within each cross-validation, two folds were used for training and the remaining fold for validation. Since there were significantly more long-survival cases (patients with DFS > 5 years, $n_p = 470$) than short-survival cases (patients with DFS ≤ 5 years, $n_N = 261$), in each cross-validation, we randomly selected the same number of long- and short-survival cases from the two-fold data to compose the training data, in order to avoid the adverse effect of unbalanced training data on the training process. To train the IGNN and IGNNE models, the Xavier initialization was used for all the layers. An adaptive moment estimation (ADAM) optimizer was adopted to learn model parameters with a batch size of 16 and initial learning rate of 0.01. The maximum training iteration were set to 325 for the IGNN model and 59 for the IGNNE model. Due to the difficulty to augment a graph dataset as a traditional image dataset, two strategies were used to prevent overfitting: first, dropout layers ($P_{\text{drop}} = 0.1$) were applied before graph convolutional layers during the training phase, and normalized layer were applied before the full connection layer. Second, an adaptive stopping scheduler (Astopper) was developed to automatically select a suitable epoch for interrupting training process. Specifically, three checkpoints were set up across the training process, and the learning rate and weight decay rate of ADAM optimizer would be updated according to the change of training loss at Checkpoint1 or Checkpoint2, respectively. Starting from Checkpoint3 until the preset maximum iteration, the epoch with the lowest training loss would be called back as the best time to interrupt training and freeze model parameters. All results from single cross-validation steps were combined as the model-specific validation results for the training cohort.

In the external validation process, the prognostic models with the same architecture as in the pre-validation were retrained with full-scale cases from the training cohort and then evaluated on both the training and validation cohorts. For IGNN (IGNNE), model parameters were again initialized by the Xavier distribution, and optimized using ADAM optimizer with a batch size of 16(128) and an initial learning rate of 0.005. The maximum training iteration were set to 656 for IGNN and 1000 for IGNNE. During training, the best number of epochs to interrupt training was determined by Astopper under the same strategy as in the pre-validation process.

**Interpretability**. To demonstrate how IGNN revealed the implicit prognostic interactions among TACS features, we trained IGNN based on the training cohort without hyperparameter tuning. Specifically, the model parameters were randomly initialized and optimized using ADAM optimizer with a batch size of 16 and a learning rate of 0.01. During training, instead of choosing the stopping moment by Astopper, the training process was randomly interrupted at the 558th epoch, and

the output feature vectors of each graph convolutional layer were extracted as the observed model responses.

To quantify the relative importance of different TACS features within patient-specific TACS graph data fed into IGNN, we introduced integrated gradient (IG), a *post hoc* gradient-based feature imputation method that attributes the model prediction to their inputs with different contributing factors. For a straight-line path from the input $\mathbf{F}$ to the associated baseline $\mathbf{F}'$, IG is defined as the path integral of the gradients along the straight-line path from $\mathbf{F}$ to $\mathbf{F}'$, which can be efficiently approximated numerically. In practice, IG for the $i$th element of $\mathbf{F}$ is computed following numerical Riemman approximation

$$\text{IG}_i(\mathbf{F}) := \frac{1}{m}(\mathbf{F}_i - \mathbf{F}_i') \sum_{k=1}^{m} \frac{\partial \text{IGNN}(\mathbf{F}' + \frac{k}{m}(\mathbf{F} - \mathbf{F}'))}{\partial \mathbf{F}_i} \qquad (11)$$

where $\mathbf{F}$ is the initial node feature matrix of graph data, $\mathbf{F}'$ is the starting point to calculate $\text{IG}_i(\mathbf{F})$ which is set to an all-zero matrix, $\text{IGNN}(\cdot)$ is the IGNN prediction, $\frac{\partial \text{IGNN}(\cdot)}{\partial \mathbf{F}_i}$ is the gradient of IGNN prediction function relative to the $i$th element of $\mathbf{F}$, $k$ is the scaled feature perturbation constant, and $m$ is the number of steps in the Riemann approximation of the integral. With the patient-specific TACS graph data as input, IG reveals the relative prognostic importance of different TACS features at the ROI level.

Training and validation of IGNN and IGNNE models were implemented using PyTorch (version 1.6.0) framework and Torch-Geometric library (version 1.6.1) with Windows 10 operating system on a computer equipped with one NVIDIA GTX1080 Ti GPU and one INTEL Core i7-6700K CPU @ 4.0 GHz.

**Statistical analysis**. The Kaplan-Meier and two-sided log-rank tests were used to plot and compare the survival curves with dichotomized predictive score from the prognostic models, which provided risk stratification assignment. The HRs of different risk groups in survival analysis and the association of clinical factors with patient DFS were measured using Cox proportional hazards regression analysis. Time-dependent AUC curves and corresponding iAUC (the measure of prognosis appropriate for censored time-to-event data) were used to assess the performance for DFS prediction of different prognostic models. The receiver operating characteristic (ROC) curves and associated AUCs were adopted to reflect the discriminatory accuracy of the prognostic models to predict 5-year DFS rate. The relative strengths of multiple prognostic factors to predict DFS was assessed using the Chi-squared test. The subgroups were compared using two-sided unpaired t-test with no adjustments made for multiple comparisons. All hypothesis analyzed with a two-sided $p$ value < 0.05 were treated as statistically significant. The 95% CIs were calculated by percentile method while t-SNE technology was used for dimensionality reduction and visual analysis of high-dimensional features. Data and statistical analyses were performed using R software (version 3.6.0). The Cox Ridge regression adopted by TACS model was performed using the "glmnet" package and the Multivariate Cox proportional hazard regression adopted by Nomogram model was performed using the "survival" package.

**Reporting summary**. Further information on research design is available in the Nature Research Reporting Summary linked to this article.

## Data availability

Raw data (whole-slide images and corresponding multiphoton images) supporting the results of this study are not publicly available due to institutional permission through IRB approval. Please email all requests for academic use of the raw data to the corresponding authors [L.L., J.C. or H.T.]. The requests will be evaluated for intellectual property or patient privacy obligations according to institutional and departmental policies. Other raw and processed data that include TACS, clinical, and follow-up information of patients are publicly available within the article, supplementary information and at [https://github.com/qldqq1984/IGNN/tree/main/experiments/Patients_Information/DataSets_995] and the raw data for all figures and tables are provided at [https://github.com/qldqq1984/IGNN/tree/main/Source%20Data]. Source data are provided with this paper.

## Code availability

The custom code related to the training and evaluation of all the prognostic models are publicly available with a detailed guide at [https://github.com/qldqq1984/IGNN]. To maximize the reproducibility, the analytical procedures with R code for Source Data are also provided to reproduce the experimental results and displayable items. These procedures have been deposited in specific folders for all figures/tables within the Source_Data_analysis folder, which can be accessed from [https://github.com/qldqq1984/IGNN/tree/main/Source_Data_analysis]. Source code has also been placed on the Zenodo platform [https://doi.org/10.5281/zenodo.6808920][48].

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

## Acknowledgements

We would like to thank Wenjiao Ren, Tingfeng Shen, Wei Wang, Zhong Chen, Xiwen Chen, Ye Fang, Xiahui Han, Na Fang, Xinxing Huang and Zhijun Li for their support in the data acquisition of multiphoton images. This work was supported by the National Natural Science Foundation of China (Grant Nos. 82171991, 82172800, 61972187), Fujian Major Scientific and Technological Special Project for "Social Development" (No. 2020YZ016002), Natural Science Foundation of Fujian Province (Nos. 2019J01269, 2019J01761, 2019J01060044, 2020J011008, 2020J01839), Joint Funds for the Innovation of Science and Technology of Fujian Province (2017Y9038, 2019Y9101), and the special Funds of the Central Government Guiding Local Science and Technology Development (No. 2020L3008). Also, this work was supported, in part, by grants from the National Institutes of Health, U.S. Department of Health and Human Services (R01 CA241618 and R41 GM139528 to H.T.).

## Author contributions

L.Q., L.L., J.C., and H.T. conceived the idea. L.Q., C.W., L.L., Q.W., X.L., and H.T. performed the analysis and wrote the manuscript. D.K., W.G., F.F., and Q.Z. collected and prepared samples. G.X., J.H., and L.Z. performed biological experiments. L.Q., D.K., L.L., J.C., and H.T. obtained funding for this research.

## Competing interests

The authors declare no competing interests.
