## [Peer Review file · Nature Communications]

REVIEWER COMMENTS

Reviewer #1 (Remarks to the Author): Expert in digital pathology, biomarkers, and deep learning

In this study, the authors address the problem of intratumor heterogeneity of biomarkers in breast cancer. They claim that the standard clinical workflow homogenizes these biomarkers in space (e.g. ER status in breast cancer) and their new method allows to extract an extra bit of information. Indeed, this is an interesting research question that could ultimately benefit patient stratification. The figures are very nice and look professional.

However, although the authors aim very high and broad, the actual work is very narrow. They use a very specific imaging technology - multiphoton microscopy MPM - as an input to a novel computational method. MPM is not being used in the clinic and it is unclear how their algorithm could be used in the real world in the light of this limitation.

The language is very convoluted and should be made more understandable to the average reader. The authors seem to massively inflate their findings by using a lot of technical terms which do not contribute to the understandability of the article. While I do understand that graph neural networks can help to uncover meaningful patterns in spatial data, the experimental design is not clearly understandable to me and this may be due to the wording. What makes this article particularly confusing is that the authors verbally jump back and forth between their very narrow domain of research and very broad and general claims.

As a reviewer, I have received a version of the source code for evaluation. I was not able to run the full code due to time limitations and the very unstructured format of the code, but I have had a close look at the source code. First of all, I would like to commend the authors on providing the source code. Nevertheless, there are a few major problems with this. The 'code availability' section of the methods does not state that the source code will be made available. This has to be corrected. Second, making the code available as a compressed file as a supplement is not best practice. 90% of all similar papers upload the code to a Github repository (or similar). In addition, in order to prevent future changes, an unchangeable checkpoint of the code should be made available on Zenodo or a similar platform.

Regarding the code itself, the documentation in the readme file is very unprofessional in terms of explanation and in terms of layout. This should be

improved. Ideally, a documentation will be made available in Markdown format on Github. The Python and R files themselves are very unstructured and almost no useful comments are available. In particular, Python best practices highly recommend to use docstrings in all functions. My impression from the code is that this will be almost unusable by other researchers in its present form. Significant work has to be invested to make the code understandable, robust and reusable.

Reviewer #2 (Remarks to the Author): Expert in computational pathology and breast cancer

Key results:

This paper adapts graph neural network to predict disease-free survival in breast cancer by inferring the interactions among tumor-associated collagen signatures in multiple regions. The presented IGNN and IGNN-E model result in better prognosis performance than the TACS1-8 model which does not consider biomarker-biomarker interactions. An important contribution of this framework is the interpretability, made possible by modelling the interactions between TACS-associated regions with graphs, which can produce valuable clinical insight.

Clarity and context:

Overall, the paper is well-written, very nicely structured, and easy to follow. This is a rigorous and substantial piece of work, showing the importance of analysing the interactions between already known biomarkers rather than the discovery of new ones for to understanding and treating cancer. It provides ample and well-presented arguments, and the numerous figures support the information very clearly. Another strength lies in the availability of different data for training and validation purposes.

Validity:

The model was extensively validated in different categories and showed promising performance as compared to the pre-established model.

Originality and significance:

This work presents a unique approach to leverage the regional parameters obtained by multi-region sampling, with the potential to incorporate different types of biomarkers. The graph is constructed using biologically meaningful patterns defined with domain knowledge, which results in a more computationally efficient and interpretable network than models based on highly variable cell-cell interactions.

The computational framework, based on recent advances in GNN architectures, is innovative in terms of architecture assembly, and with respect to the application on ECM specific biomarkers interaction modelling. This is an important and relevant contribution reflecting the need to incorporate the interactions between ECM biomarkers into disease prognosis, with a view to integrate this into clinical practice. The post hoc interpretations are of particular interest, showcasing the remarkable power of GNNs to learn relevant node embeddings that capture interpretable prognostic interactions between the biomarkers.

Data & methodology:

The description of dataset is comprehensive and informative. Interestingly, the authors exploit intratumor heterogeneity upon constructing the prognostic model, but not directly the biomarker diversity, which is deemed to be controversial for a prognostic tool. Even so, the choice of ROI (number, position, size - choices which are not well explained in this study) is not clearly justified, and thus reflects an arbitrary “level” of intra tumor heterogeneity i.e. changing the position of one region would completely change the resulting graph in terms of adjacencies. Given that one slide/patient is represented through a graph connecting the sampled regions/slide, a discussion focused on the necessary number/size of regions to construct a representative graph, would have been helpful, with a view to propose it as a generalizable method. In other words, while it is true that the proven prognostic value shows remarkable potential, it is not yet clear how the size and numbers of ROIs might affect the predictive performance.

Another observation is that the authors are not clear on the proven benefit of using one IGNN-E (i.e. extended model incorporating clinical information) vs IGNN, since in most analyses they are taken together against TACS-based model. It would be helpful to understand how incorporating clinical scores into the decision graph-based model was relevant/different for the prognosis.

Although the pipelines are detailed and very well described, we encourage the authors to publish the codes, which would facilitate the method reproducibility and testing.

Appropriate use of statistics and treatment of uncertainties:

Error bars of boxplots are not clearly defined in the figure legends.

Conclusions:

Pros and cons of the study are nicely discussed and the scope for future development is provided.

Suggested improvements:

Please refer to previous section (i.e. Data & Methodology) for additional suggestions regarding the methodology

1. Please provide references when mentioning the roles of TACS in tumor development.
2. In the post hoc interpretation section, the author suggests that TACS2,3,7,8 tend to affect survival by interacting with other TACSs. Is this likely because TACS2,3,7,8 are relatively rare in the samples so their patterns are not as apparent as the dominant TACS? For example, TACS8 percentage seems to be higher in cluster1 compared to the other two clusters.

Response to reviewers' comments

Reviewer #1 (Remarks to the Author): Expert in digital pathology, biomarkers, and deep learning

In this study, the authors address the problem of intratumor heterogeneity of biomarkers in breast cancer. They claim that the standard clinical workflow homogenizes these biomarkers in space (e.g. ER status in breast cancer) and their new method allows to extract an extra bit of information. Indeed, this is an interesting research question that could ultimately benefit patient stratification. The figures are very nice and look professional.

We thank Reviewer #1 for overall positive comments on our study. We believe more efforts along this direction will pave the way for the clinical translation of novel optical technologies.

However, although the authors aim very high and broad, the actual work is very narrow. They use a very specific imaging technology - multiphoton microscopy MPM - as an input to a novel computational method. MPM is not being used in the clinic and it is unclear how their algorithm could be used in the real world in the light of this limitation.

In our opinion, the scope of this work is not narrow. First, multiphoton microscopy (MPM) is not a highly specific imaging technology, as would be the case two decades ago. With the maturation of ultrafast laser technology, sensitive photodetectors, reliable scanners, and DIY microscope components, MPM has been widely explored in pathology in a research setting. Second, the clinical translation of novel optical imaging technologies (such as MPM) is not limited by their technical flaws in comparison to standard (H&E) histology, but by the adoption of the pathologists' view for diagnosis purposes (see J. Biomed. Opt. 12, 051801, 2007 "Validation of novel optical imaging technologies: the pathologists' view"). In this sense, our work reflects a systematic effort to adopt the pathologists' view on novel optical imaging markers via a machine learning algorithm. Third, the model developed in this study is not limited to MPM technology as long as the biomarkers possess spatial heterogeneity.

Throughout the revised manuscript, we have deemphasized MPM and put this technology into a broad context of novel optical imaging technologies, and thus avoided the unintended impression that our model would work for MPM only. In particular, in the end of Introduction section, we have discussed the potential application of our method in a clinical setting, with the assistance of a new supplementary table (Table S1). We state that "Our distance-less IGNN model (see below) paves the way for new (non-imaging) tissue assessment technologies beyond traditional histopathology of FFPE sections (Table S1)".

The language is very convoluted and should be made more understandable to the average reader. The authors seem to massively inflate their findings by using a lot of technical terms which do not contribute to the understandability of the article. While I do understand that graph neural networks can help to uncover meaningful patterns in spatial data, the experimental design is not clearly understandable to me and this may be due to the wording. What makes this article particularly confusing is that the authors verbally jump back and forth between their very narrow domain of research and very broad and general claims.

We believe our choices of language and terminology are relatively typical in optical imaging, histology, and machine learning communities (we did not create or invent new terms other than IGNN). Also, the major findings of this study (Figs. 2-4) arise from standard data analysis, which is not relevant to specific technical terms in use. Thus, this issue of understandability in our original manuscript may be due to insufficient connection between the “specific” research in multiphoton microscopy (narrow domain) with the general claim of improved cancer prognosis (broad scope).

To improve the understandability and readability, we have made four major changes in our revised manuscript. First, in the end of Introduction section, we explicitly point out that “Although we demonstrate the benefits of this model using imaging biomarkers of TACS1-8 and MPM, the model itself does not depend on the nature of the biomarkers (morphology vs. molecular) or associated technologies (H&E, MPM, other novel optical imaging technologies), as long as the biomarkers possess spatial heterogeneity (which is broadly valid due to the well-documented intratumor heterogeneity)”. Thus, the model is generally applicable to diverse technologies and biomarkers summarized in a new supplementary table (Table S1), and the perceived terminological disconnection between narrow research and broad claim is avoided. Second, technical terms, such as TACS, SHG, TPEF, and GRU, are defined when they first occur along with the corresponding references (Refs. 14,15,16,19). Third, at the beginning of the Results section, we provide more details on our experimental design. We explicitly point out that “One section was stained with H&E for whole slide imaging, in which a pathologist confirmed the presence of tumor cells and their borders. Dependent on the size of tumor area for sufficient sampling, several (4-20) ~2.8-mm-sized non-overlapping regions of interest (ROI) were located mainly at the tumor invasive front and then labeled (numbered) in the H&E images. The other (unstained) section was deparaffinized by alcohol and xylene to collect label-free dual-modal MPM of second harmonic generation (SHG) and two-photon excited (intrinsic) fluorescence (TPEF) images for all labeled ROIs”. Fourth, we have provided more details on the training/validation and interpretability of our IGNN models in revised Methods as well as Supplementary Information (Table S6 and Fig. S4).

Table S6. Performance to predict 5-year DFS rates of patients in training cohort by four prognostic models.

Pre-validation						
Models	Training cohort (n=731)					
	AUC _{risk}	Sens	Spec	PPV	NPV	Acc
TACS1-8	0.766	0.759	0.770	0.647	0.852	0.766
IGNN	0.791	0.824	0.757	0.654	0.886	0.781
Nomogram	0.775	0.793	0.757	0.645	0.868	0.770
IGNN-E	0.819	0.847	0.792	0.693	0.903	0.811

*AUC_{risk}: area under the receiver operating characteristic curve according to the prognostic risk; **Sens**: sensitivity; **Spec**: specificity; **PPV**: positive predict value; **NPV**: negative predict value; **Acc**: accuracy.

Figure S4. Performance of four prognostic models in pre-validation. (a) Kaplan-Meier survival analysis of patients stratified into high risk group (red line) and low risk group (blue line) by different prognostic models (two-sided log-rank test to determine significance p). (b) Distribution of C-index (top) and iAUC panel (middle), and ROC curves and AUCs for 5-year DFS rate of prognostic risk predicted by different models (bottom). (c) Relative contributions of prognostic biomarkers in predicting DFS in training cohort according to multivariate Cox proportional hazard regression analysis (Chi-squared test to determine p).

As a reviewer, I have received a version of the source code for evaluation. I was not able to run the full code due to time limitations and the very unstructured format of the code, but I have had a close look at the source code. First of all, I would like to commend the authors on providing the source code. Nevertheless, there are a few major problems with this. The 'code availability' section of the methods does not state that the source code will be made available. This has to be corrected. Second, making the code available as a compressed file as a supplement is not best practice. 90% of all similar papers upload the code to a Github repository (or similar). In addition, in order to prevent future changes, an unchangeable checkpoint of the code should be made available on Zenodo or a similar platform. Regarding the code itself, the documentation in the readme file is very unprofessional in terms of explanation and in terms of layout. This should be improved. Ideally, a documentation will be made available in Markdown format on Github. The Python and R files themselves are very unstructured and almost no useful comments are available. In particular, Python best practices highly recommend to use docstrings in all functions. My impression from the code is that this will be almost unusable by other researchers in its present form. Significant work has to be invested to make the code understandable, robust and reusable.

We appreciate these comments to improve the usability of our source code and have followed the corresponding suggestions. In the revised manuscript, the 'code availability' section has stated that the source code will be made available. We have provided the raw data including TACS, clinical, and follow-up information of patients and the Source Code related to the training and evaluation of the prognostic models. In addition, parameters from the well-trained models are provided to reproduce the model performance and experimental results presented in this paper, and we also provided a template program that allows users to train and evaluate IGNN (IGNNE) models using custom data from scratch. We have significantly revised the readme file to improve the understandability. All the raw data and Source Code are publicly available with a detailed guide (README.md) at [\[https://github.com/qldqq1984/IGNN\]](https://github.com/qldqq1984/IGNN). The Source code has also been placed on the Zenodo platform [\[https://doi.org/10.5281/zenodo.5689267\]](https://doi.org/10.5281/zenodo.5689267). The source code with python and R has been extensively tested in different platforms while the necessary comments with docstrings have been added in the code snippets and functions to make the code robust and understandable.

Reviewer #2 (Remarks to the Author): Expert in computational pathology and breast cancer

Key results:

This paper adapts graph neural network to predict disease-free survival in breast cancer by inferring the interactions among tumor-associated collagen signatures in multiple regions. The presented IGNN and IGNN-E model result in better prognosis performance than the TACS1-8 model which does not consider biomarker-biomarker interactions. An important contribution of this framework is the interpretability, made possible by modelling the interactions between TACS-associated regions with graphs, which can produce valuable clinical insight.

Clarity and context:

Overall, the paper is well-written, very nicely structured, and easy to follow. This is a rigorous and substantial piece of work, showing the importance of analysing the interactions between already known biomarkers rather than the discovery of new ones for to understanding and treating cancer. It provides ample and well-presented arguments, and the numerous figures support the information very clearly. Another strength lies in the availability of different data for training and validation purposes.

Validity:

The model was extensively validated in different categories and showed promising performance as compared to the pre-established model.

Originality and significance:

This work presents a unique approach to leverage the regional parameters obtained by multi-region sampling, with the potential to incorporate different types of biomarkers. The graph is constructed using biologically meaningful patterns defined with domain knowledge, which results in a more computationally efficient and interpretable network than models based on highly variable cell-cell interactions.

The computational framework, based on recent advances in GNN architectures, is innovative in terms of architecture assembly, and with respect to the application on ECM specific biomarkers interaction modelling. This is an important and relevant contribution reflecting the need to incorporate the interactions between ECM biomarkers into disease prognosis, with a view to integrate this into clinical practice. The post hoc interpretations are of particular interest, showcasing the remarkable power of GNNs to learn relevant node embeddings that capture interpretable prognostic interactions between the biomarkers.

Data & methodology:

The description of dataset is comprehensive and informative. Interestingly, the authors exploit intratumor heterogeneity upon constructing the prognostic model, but not directly the biomarker diversity, which is deemed to be controversial for a prognostic tool.

We thank Reviewer #2 for overall positive comments on our study. We believe our study represents a novel way to exploit the benefits of intratumor heterogeneity (as appreciated by this reviewer).

Even so, the choice of ROI (number, position, size - choices which are not well explained in this study) is not clearly justified, and thus reflects an arbitrary “level” of intra tumor heterogeneity i.e. changing the position of one region would completely change the resulting graph in terms of adjacencies. Given that one slide/patient is represented though a graph connecting the sampled regions/slide, a discussion focused on the necessary number/size of regions to construct a representative graph, would have been helpful, with a view to propose it as a generalizable method. In other words, while it is true that the proven prognostic value shows remarkable potential, it is not yet clear how the size and numbers of ROIs might affect the predictive performance.

We thank Reviewer #2 for this helpful suggestion. In the revised manuscript, we justify our choice in the Discussion section (2nd paragraph) by explicitly stating “As an early attempt to exploit the benefits of intratumor heterogeneity, we have chosen a size of ~2.8 mm for ROIs, each of which produces ~1.3 distinct features of TACS1-8 (Table S9). For this size, representative sampling of the tumor invasion front of a typical histological section requires 4-20 ROIs, depending on the dimension of the tumor area. The number of ROIs seem adequate because random removal of 20% of ROIs for individual patients results in rather small inconsistency in prognosis (Table S10). Thus, it is unlikely that our particular choice of ROIs (number and position) introduces an arbitrary level of intratumor heterogeneity. However, our combination of ROI size, sampling site (invasion front), and number of ROIs may not be optimal to attain the full potential of the IGNN model. Future efforts to balance the trade-off between ROI size (content of TCASs in one region) and ROI number (content of TACS-TACS interaction among regions), and the trade-off of overall TACS content versus cost, may yield better overall performance in a clinical setting. The extension of TACS content from tumor invasion front to tumor center may further improve the performance”.

Table S9. TACS-containing ROIs from all patients under study ($n = 995$).

Total number of ROIs	ROIs with 1 TACS features	ROIs with 2 TACS features	ROIs with 3 TACS features	ROIs with ≥ 4 TACS features
7424	5357	1781	261	25
percentage	72.2%	24.0%	3.51%	0.34%

Table S10. Effect of random removal of 20% of ROIs on IGNN prognosis of individual patients in the validation cohort with >6 ROIs ($n = 212$).

Trial	Consistency (high risk)	Consistency (low risk)
1	0.963	0.969
2	0.963	0.977
3	0.951	0.969
4	0.975	0.969
5	0.975	0.969
6	0.975	0.969
7	0.975	0.969
8	0.963	0.977
9	0.951	0.969
10	0.951	0.969
11	0.951	0.977
12	0.963	0.977
13	0.963	0.969
14	0.963	0.969
15	0.963	0.969
16	0.963	0.969
17	0.963	0.969
18	0.951	0.969
19	0.975	0.969
20	0.963	0.969

Note: To evaluate the impact of potential variability in the number and distribution of selected ROIs on the risk stratification ability of IGNN model, we randomly removed 20% ROIs from the sample and repeat the prediction using the well-trained IGNN model. The new prediction remains highly consistent with the original prediction based on all ROIs.

Another observation is that the authors are not clear on the proven benefit of using one IGNN-E (i.e. extended model incorporating clinical information) vs IGNN, since in most analyses they are taken together against TACS-based model. It would be helpful to understand how incorporating clinical scores into the decision graph-based model was relevant/different for the prognosis.

We agree that the discussion on the IGNN-E and clinical information is somewhat fragmented in our original manuscript. In the revised manuscript, we explicitly state that “With no new biomarker beyond TACS1-8 (8 in situ biomarkers), our IGNN recovered this differential prognostic value from regional interactions among TACS1-8, which approximated the differential prognostic value from the combination of routine biomarkers (i.e. the Nomogram over TACS1-8 model) (Fig. 2a) particularly for patients with a small (<2 cm) tumor (Fig. 2b). These two differential prognostic values were largely additive from the TACS1-8 to IGNN-E model (Figs. 2a-2b), indicating rather independent prognosis of these regional interactions from the routine prognostic biomarkers. Thus, the combined IGNN score and traditional clinicopathological factors synergistically improve the performance of cancer prognosis”. Thus, the benefit to include clinical information is clarified.

Although the pipelines are detailed and very well described, we encourage the authors to publish the codes, which would facilitate the method reproducibility and testing.

We have made our raw data and source code publicly available by depositing it in a suitable repository (Github), as suggested by this reviewer.

Appropriate use of statistics and treatment of uncertainties:

Error bars of boxplots are not clearly defined in the figure legends.

Clear definitions and statistical method statements for all displayable items have been implemented in revised figure legends.

Conclusions:

Pros and cons of the study are nicely discussed and the scope for future development is provided.

Suggested improvements:

Please refer to previous section (i.e. Data & Methodology) for additional suggestions regarding the methodology

1. Please provide references when mentioning the roles of TACS in tumor development.

This has been done by adding Ref. 14 and Ref. 15 (review papers).

2. In the post hoc interpretation section, the author suggests that TACS2,3,7,8 tend to affect survival by interacting with other TACSs. Is this likely because TACS2,3,7,8 are relatively rare in the samples so their patterns are not as apparent as the dominant TACS? For example, TACS8 percentage seems to be higher in cluster1 compared to the other two clusters.

We perform additional Analysis of similarities (ANOSIM) in Fig. S9 on ROIs with rare TACS features (TACS2,3,7,8) and clusters with dominant TACS features (TACS1,4,5,6). In the revised manuscript, we explicitly state that “no clusters with high correlation between intra-cluster regions contain a high percentage of regions with TACS2, TACS3, TACS7, and TACS8, which may be due to the relative rarity of TACS2,3,7,8 in the samples. Additionally, the proportions of TACS2,3,7,8 are unevenly distributed across different clusters (Figs. 4a-4c, middle), and ANOSIM shows that ROIs with TACS3 or TACS8 correlate more with Cluster 1 than other clusters while ROIs with TACS2 (TACS7) correlate more with Cluster 3 (Cluster 2) (Fig. S9). Considering collagen morphologies, we hypothesize that TACS2,3,7,8 affect survival by interacting with other TACSs. For example, sparsely distributed collagen fibers at the tumor invasion front (TACS8) is conducive to TACS5,6 that enables tumor cell migration, while densely distributed collagen fibers at the tumor invasion front (TACS7) is conducive to TACS4 that restricts tumor cell migration”.

Figure S9. ANOSIM assessment on ROIs with rare TACS features (TACS2,3,7,8) and clusters with dominant TACS features (TACS1,4,5,6). An R value close to 1.0 suggests dissimilarity between the compared regions while an R values below 0 suggest that dissimilarities are greater within the compared regions than between the compared regions. The position of the observed R value over the null distribution is used to assess the significance of the R statistic (p value).

REVIEWER COMMENTS

Reviewer #1 (Remarks to the Author):

I would like to thank the authors for their detailed response. However, my main comment was that the authors empirically show the benefits of their method for MPM only. MPM is a research-only technology and is not used at all in routine diagnostics. The authors claim that the new data analysis method works also for routine image data, but this is not empirically shown.

In particular, in their response the authors state that: "Throughout the revised manuscript, we have deemphasized MPM and put this technology into a broad context of novel optical imaging technologies, and thus avoided the unintended impression that our model would work for MPM only."

This is exactly the wrong way to go, in my opinion. The claim that the method is applicable to any type of data cannot be made based on empirical data showing its applicability only to MPM.

The last sentence of the abstract claims that this study shows "a cost-effective route to precision medicine" - I do not agree. In my point of view this is a highly technically interesting study, but it is remote from clinical application.

Reviewer #2 (Remarks to the Author):

The authors perform additional ANOSIM analysis for rare TACS features. Fig S9 a, b are missing the title. The conclusion 'ANOSIM shows that ROIs with TACS3 or TACS8 correlate more with Cluster 1 than other clusters while ROIs with TACS2 (TACS7) correlate more with Cluster 3 (Cluster 2).' Is not obvious from the figure. All TACS2,3,7,8 appear to share more similarities with Cluster1 than other clusters. The 'between clusters' and 'within clusters' labels are confusing. It's also not clear how the correlation can be inferred from the ANOSIM R value.

Reviewer #3 (Remarks to the Author):

In this well written manuscript, the authors aimed to demonstrate that TACS (tumour-associated collagen signature) might be used to determine prognosis, using an intratumor graph neural network. Further, addressing the problem of intratumor heterogeneity of routinely breast cancer biomarkers and improvement of prognosis stratification. However, some points may be of concern:

01. The authors used surgical specimens of adjuvant therapy patients and the histology was only of a special type of breast carcinoma. Nowadays, it is more common with neoadjuvant surgical specimens (doi: 10.1177/1178223419829072), so only biopsies are free of treatment. How could this be handled? In patients with treatment, fibrosis will certainly be different, including those with a complete response from those with a partial response.

02. The authors are clear and persuasive in “[...]our IGNN recovered this differential prognostic value from regional interactions among TACS1-8, which approximated the differential prognostic value from the combination of routine biomarkers [...], particularly for patients with a small (< 2 cm) tumor[...]” and “the observed differential prognostic value of patients with a small \leq 2 cm tumor approximated that from the combination of routine biomarkers, whereas patients with moderate 2-5 cm or large > 5 cm tumour obtained a less significant result[...]”. In cases of multifocal tumour, which foci were used to establish the invasion front and ROI chosen? If one focus is smaller than 2 cm and the other is larger, what would be the way to evaluate?

In the article that showed the development of TACS 1-8 (doi:10.7150/thno.55921), TACS was an independent prognostic factor along tumour size, lymph node status and molecular subtype. Is there any justification for being distinguished in your work?

03. Neoplastic cells or other cells do not provide second-harmonic generation signal. Among the main emitters, collagen stands out. In the ROI selections, was the stroma:cellularity ratio evaluated?

If the choice of an ROI is a high stroma:cellularity ratio, we would have a large amount of collagen fibers to be evaluated, while a low stroma:cellularity ratio, we would have a low amount of collagen to be evaluated. In Figure 1d, this is easily seen from ROI 1 to ROI 8. Wouldn't that change the results since we would change the vectors? I believe it is different from the number and size of selected ROIs, already discussed so well in the first review

04. A curiosity, since the evaluation was performed on SHG images, why the option to perform microscopy on the unstained slide was made. Neither eosin nor hematoxylin alters the SHG signal. Wouldn't it be more faithful to what the pathologist selected on the slides?

05. I found it interesting how they compensated for certain findings during the development of the network. However, it was not so clear how samples that presented extremely variable TACS were

weighted, that is, in the same sample they present TACS-associated to good prognosis ROIs and TACS-associated to bad prognosis ROIs. I think that would be an interesting point to make clear.

06. Different histological subtypes of breast cancer present different collagen fibres architecture (doi: 10.1038/s41598-019-44156-9). Therefore, you should be careful with "[...] can be expanded to other reported in situ biomarkers, e.g. histological type[...]"

In short, the great challenge for this manuscript is still to demonstrate reproducibility, even using artificial intelligence. However, despite the limitations, it is still an original manuscript and one that set out to develop something interesting in an extremely heterogeneous disease.

Reviewer #1 (Remarks to the Author):

I would like to thank the authors for their detailed response. However, my main comment was that the authors empirically show the benefits of their method for MPM only. MPM is a research-only technology and is not used at all in routine diagnostics. The authors claim that the new data analysis method works also for routine image data, but this is not empirically shown.

We also thank the reviewer for additional efforts to improve our presentation. It is true that MPM is a research-only technology not used in routine diagnostics and our data analysis method has not been demonstrated for non-MPM image data. Thus, we tone down the claim on the clinical implications of our data analysis method.

In particular, in the Introduction section, we remove "Our distance-less IGNN model (see below) paves the way for new (non-imaging) tissue assessment technologies beyond traditional histopathology of FFPE sections (Table S1). Although we demonstrate the benefits of this model using imaging biomarkers of TACS1-8 and MPM, the model itself does not depend on the nature of the biomarkers (morphology vs. molecular) or associated technologies (H&E, IHC, MPM, etc.), as long as the biomarkers possess spatial heterogeneity (which is broadly valid due to the well-documented intratumor heterogeneity)". Instead, we state that "Our distance-less IGNN model based on MPM may motivate the development of similar models based on other imaging and non-imaging tissue assessment technologies (Table S1), as long as the biomarkers possess spatial heterogeneity".

Also, in the Discussion section, we remove the statement of "Although our IGNN model is demonstrated using MPM imaging, cost-effective multiregion micro-spectroscopy (molecular sensing) may be more attractive for the potential application of this model in a clinical setting (Table S1). With the IGNN model of relevant spectroscopic biomarkers, this model may be expanded from sectioned/stained tissue disease prognosis to live-tissue intraoperative cancer diagnosis (Table S1). The integrative omics technology⁴⁷ may also benefit from the IGNN model once the cost of multiregion omics is lowered considerably".

In particular, in their response the authors state that: "Throughout the revised manuscript, we have deemphasized MPM and put this technology into a broad context of novel optical imaging technologies, and thus avoided the unintended impression that our model would work for MPM only." This is exactly the wrong way to go, in my opinion. The claim that the method is applicable to any type of data cannot be made based on empirical data showing its applicability only to MPM.

These points are similar to the comment above. In the revised manuscript, we have reemphasized MPM in our model development and avoided the impression that our data analysis method is readily generalizable to non-MPM technologies/data.

The last sentence of the abstract claims that this study shows "a cost-effective route to precision medicine" - I do not agree. In my point of view this is a highly technically interesting study, but it is remote from clinical application.

We thank the reviewer for the overall assessment of our work as “a highly technically interesting study”. We agree that the technique developed in this work has a long way toward clinical application. In the revised manuscript, we do not claim “a cost-effective route to precision medicine” but simply state that “Our study demonstrates an alternative route to cancer prognosis...”.

The perspective clinical application of this study will depend on the clinical translation of MPM. There have been portable MPM microscopes reported for intraoperative imaging (Intraoperative visualization of the tumor microenvironment and quantification of extracellular vesicles by label-free nonlinear imaging. *Sci. Adv.* 4:eaau5603, 2018) and there have been commercial MPM microscopes intended for histopathology (e.g. <http://www.femtodiagnostics.nl/>; <http://www.livebiopsy.com/>).

Reviewer #2 (Remarks to the Author):

The authors perform additional ANOSIM analysis for rare TACS features. Fig S9 a, b are missing the title. The conclusion 'ANOSIM shows that ROIs with TACS3 or TACS8 correlate more with Cluster 1 than other clusters while ROIs with TACS2 (TACS7) correlate more with Cluster 3 (Cluster 2).' Is not obvious from the figure. All TACS2,3,7,8 appear to share more similarities with Cluster1 than other clusters. The 'between clusters' and 'within clusters' labels are confusing. It's also not clear how the correlation can be inferred from the ANOSIM R value.

We thank the reviewer for additional effort to improve the readability of this manuscript. The ANOSIM analysis intends to assess the similarity between the group consisting of ROIs with rare TACS features (TACS2,3,7,8) and the group consisting of ROIs with dominant TACS features (TACS1,4,5,6). Given the matrix of Bray-Curtis rank dissimilarities between all the samples from the groups under comparison, the ANOSIM tests the null hypothesis that the similarity between groups is greater than the similarity within the groups. The test statistic R is calculated as

$$R = \frac{\bar{r}_B - \bar{r}_w}{0.25 n(n - 1)}$$

where \bar{r}_B is the mean Bray-Curtis rank dissimilarity for sample pairs from different groups, \bar{r}_w is the mean Bray-Curtis rank dissimilarity for sample pairs within the same groups, and n is the total number of samples. R ranges from -1 to 1 , wherein $R < 0$ suggests that dissimilarities are greater within groups than between groups while $R > 0$ suggests greater dissimilarities between groups than within groups. The closer the R value to 1 , the greater the dissimilarities between samples from different groups.

We have modified Figure S9 in the revised manuscript. In Fig S9a, the lowest R value ($R = 0.47$) indicates the smallest dissimilarity between Cluster3 and the group consisting of ROI samples with TACS2. In Fig S9 (b), the lowest R value ($R = 0.21$) indicates the smallest dissimilarity between Cluster1 and the group consisting of ROI samples with TACS3. In Fig S9 (c), the lowest R value ($R = 0.51$) indicates the smallest dissimilarity between Cluster2 and the group consisting of ROI samples with TACS7. In Fig S9 (d), the lowest R value ($R = 0.26$) indicates the smallest dissimilarity between Cluster1 and the group consisting of ROI samples with TACS8.

Thus, it is reasonable to retain the conclusion in the original manuscript "...ANOSIM shows that ROIs with TACS3 or TACS8 correlate more with Cluster 1 than other clusters while ROIs with TACS2 (TACS7) correlate more with Cluster 3 (Cluster 2)..."

Reviewer #3 (Remarks to the Author):

In this well written manuscript, the authors aimed to demonstrated that TACS (tumour-associated collagen signature) might be used to determinate prognosis, using an intratumor graph neural network. Further, addressing the problem of intratumor heterogeneity of routinely breast cancer biomarkers and improvement of prognosis stratification.

We thank the reviewer for appreciating the significance of this work.

However, some points may be concern:

01. The authors used surgical specimens of adjuvant therapy patients and the histology was only no special type of breast carcinoma. Nowadays, it is more common neoadjuvant surgical specimens (doi: 10.1177/1178223419829072), so only biopsies are free of treatment. How could this be handled? In patients with treatment, fibrosis will certainly be different, including those with a complete response from those with a partial response.

The reviewer raises a valid point on how to handle the patients with neoadjuvant therapy. During the timeframe of this study, we did examine some surgical specimens of neoadjuvant therapy patients and found out that the corresponding TACS features, with often dense distribution (fibrosis), differed considerably from those of adjuvant therapy patients. Thus, our model is restricted to adjuvant therapy patients (which may be considered as a limitation). A similar IGNN model may be established for neoadjuvant therapy patients if the variability of neoadjuvant therapy itself is small (which could be challenging). That said, we believe the restriction to adjuvant therapy patients is not a large limitation because it is unlikely that neoadjuvant therapy would ultimately supersede adjuvant therapy, particularly in cases of small early-detected tumors. In the hospitals involved in this study, the portions of adjuvant and neoadjuvant therapy patients are 70% and 30%, respectively.

In the end of the revised manuscript, we explicitly point out that “our IGNN model is restricted to adjuvant therapy patients, and is not applicable to neoadjuvant therapy patients whose tumors and TACS features have been perturbed by the neoadjuvant therapy⁴⁷”. We also add the reference on neoadjuvant therapy (doi: 10.1177/1178223419829072) as ref. 47.

We thank the reviewer for the hint on examining the small biopsies free of treatment (rather than large surgical specimens). Indeed, multi-core breast biopsies are amenable for the label-free imaging of this study to detect TACS features. It will be interesting to conduct this imaging on fresh biopsies so that our IGNN prognosis may be completed before either adjuvant or neoadjuvant therapy.

02. The authors are clear and persuasive in “[...]our IGNN recovered this differential prognostic value from regional interactions among TACS1-8, which approximated the differential prognostic value from the combination of routine biomarkers [...], particularly for patients with a small (< 2 cm) tumor[...]” and “the observed differential prognostic value of patients with a small \leq 2 cm tumor approximated that from the combination of routine biomarkers, whereas patients with moderate 2-5 cm or large > 5 cm tumour obtained a less significant result[...]”. In cases of multifocal tumour, which foci was used to

establish the invasion front and ROI chosen? If one focus is smaller than 2 cm and the other is larger, what would be the way to evaluate?

In the article that showed the development of TACS 1-8 (doi:10.7150/thno.55921), TACS was an independent prognostic factor along tumour size, lymph node status and molecular subtype. Is there any justification for being distinguished in your work?

We agree that there is ambiguity encountered in a patient with multifocal tumors. For this patient, standard histopathology was performed for all foci. However, for simplicity, the MPM imaging to detect TACS features was performed only on one of the foci with the largest size. It is possible that one of the foci with a smaller size may also be representative of the prognosis of this patient via our model. This feasibility requires a separate study dedicated to a large number patients with multifocal tumors. In this work involving 995 patients, only ~5% had multifocal tumors, so that we lack the statistics to investigate this feasibility. On the other hand, the small percentage of the patients with multifocal tumors suggests that our IGNN model should not be dramatically affected by this complexity.

To point out this complexity in the revised manuscript, we explicitly state in the Methods section that "Only a small portion (~5%) of patients has multifocal tumors, and in this case, the sample preparation was performed on one of the foci with the largest size."

Through multivariate Cox proportional hazards regression analysis, IGNN score has been justified as an independent prognostic factor just like TACS score (see Table S6). In the revised manuscript, we explicitly state "Just like the TACS score¹¹, the IGNN score functioned as an independent prognostic factor along with tumor size, lymph node status, and molecular subtype".

03. Neoplastic cells or other cells do not provide second-harmonic generation signal. Among the main emitters, collagen stands out. In the ROI selections, was the stroma:cellularity ratio evaluated? If the choice of an ROI is a high stroma:cellular ratio, we would have a large amount of collagen fibers to be evaluated, while a low stroma:cellular ratio, we would have a low amount of collagen to be evaluated. In Figure 1d, this is easily seen from ROI 1 to ROI 8. Wouldn't that change the results since we would change the vectors? I believe it is different from the number and size of selected ROIs, already discussed so well in the first review

For a given patient with a specific selection of ROIs, TACS features were identified manually whereas the stroma:cellularity ratio was not evaluated. We agree that this ratio is an important factor that dictates the reproducible prognosis for this patient based on intratumoral heterogeneity. In the first review, we have provided evidences on why our particular choice of ROIs (number and position) does not introduce an arbitrary level of intratumoral heterogeneity. The variation of stroma:cellularity ratio among ROIs is an even bigger challenge because simple assay to quantify this ratio is not available. Although our own attempt has attributed TACS7 (or TACS8) to a high (or low) stroma:cellularity ratio, the assessment is qualitative and not generalizable to other TACS features. To ultimately avoid the arbitrary level of intratumoral heterogeneity generated by the stroma:cellularity ratio, the tumor should be fully sampled and a convergent prognosis be demonstrated for each patient with different selections of ROIs. This would require 10-time faster in imaging speed of our multiphoton microscope

or a lower imaging resolution to attain the same throughput (~40 min/patient). We are working on the later approach because it appears that a low imaging resolution afforded by a 10x N.A. = 0.5 microscope objective (instead of 20x N.A. = 0.8 in this study) does not degrade our ability to identify TACS features at a relative large spatial scale (2.8 mm).

At the time being, we acknowledge the absent evaluation of the stroma:cellularity ratio as a limitation of our study, and explicitly point out in the end of the revised manuscript that “the insufficient sampling of a tumor with specific selection of ROIs may introduce prognosis uncertainty at individual level (but not necessarily population level) due to uncertain intratumoral heterogeneity such as stroma versus cellularity ratio”.

The reason why the uncertain stroma:cellularity ratio does not necessarily introduce prognosis uncertainty at population level is due to the statistical significance of our model that representatively sample 995 tumor slides (from 995 patients) via ~8000 ROIs. Thus, the bias in prognosis due to uncertain stroma:cellularity ratio in some patients may be canceled by that in other patients.

04. A curiosity, since the evaluation was performed on SHG images, why the option to perform microscopy on the unstained slide was made. Neither eosin nor hematoxylin alters the SHG signal. Wouldn't it be more faithful to what the pathologist selected on the slides?

There are two reasons that we chose the unstained slide: (1) the variability in H&E staining in sample preparation is avoided; and (2) the model based on the unstained slide may be readily generalized to fresh surgical tissue for fast cancer prognosis. It should be noted that the identification of TACS features require not only SHG images but also auto-fluorescence images (see Ref. 11). Recently, we begin to recognize that there are important advantages by choosing the H&E slides for SHG evaluation only (one of which is the faithfulness to what the pathologist selected on the slides, as suggested). We are working on whether our IGNN model can migrate from the unstained slide to the H&E slide via transfer learning.

05. I found it interesting how they compensated for certain findings during the development of the network. However, it was not so clear how samples that presented extremely variable TACS were weighted, that is, in the same sample they present TACS-associated to good prognosis ROIs and TACS-associated to bad prognosis ROIs. I think that would be an interesting point to make clear.

We appreciate the suggestion to improve the presentation on how the network was developed. In a heterogeneous tumor microenvironment, the prognostic contributions of TACS features are closely related to their distributions and interactions. Thus, TACS features are considered as integrated components of a personalized relational biomarker network of IGNN. As to weighted TACS features, unlike traditional models (such as ridge regression or LASSO regression) which explicitly characterize the importance of prognostic factor through a weight value, IGNN learns on the personalized graph data during training and determines the prognostic importance of TACS features by updating and eventually solidifying a series of weight parameters that directly or indirectly control feature expressiveness. In other words, the prognostic value of TACS features is not revealed by individual

weight values but by a series of correlative weight parameters learned from the intended task. We have clarified this point in the section of "Architectures of IGNN and IGNN-E" in multiple instances:

"...architecture that drives nodes to aggregate information with their neighbors along edges and perceives their interactions ..."

"...The attention weight α_{ij} are assigned to different nodes and edges according to the underlying dependencies to direct the network to the most prognostic parts of the TACS-based graph structure ..."

"...Specially, in the non-linearity mapping process for converting the initial node state

vectors $\mathbf{F} = [\mathbf{v}_1^T, \dots, \mathbf{v}_N^T]^T \in \mathbb{R}^{N \times 8}$ into higher-level node feature embeddings $\mathbf{H}^{(1)} = \text{Selu}(\mathbf{F}\mathbf{W}_f)$, the feature vectors \mathbf{F} are weighted by a learnable weight matrix $\mathbf{W}_f \in \mathbb{R}^{8 \times c^{(1)}}$ shared among all the nodes to highlight their relative importance in the $\mathbf{H}^{(1)}$..."

"...Fully connected network module weights the portions of features $\mathbf{h}^{(t)}$ with different importance level via a learnable weight matrix $\mathbf{W}_h^{(t)}$ and recombines them into more discriminative higher-level features $\mathbf{h}^{(t+1)}$ by the non-linearity mapping of $\mathbf{h}^{(t+1)} = \text{Selu}(\mathbf{h}^{(t)}\mathbf{W}_h^{(t)} + \mathbf{b}_h^{(t)})$..."

"To quantify the relative importance of different TACS features within patient-specific TACS graph data fed into IGNN, we introduced integrated gradient (IG), a post hoc gradient-based feature imputation method that attributes the model prediction to their inputs with different contributing factors. For a straight-line path from the input F to the associated baseline F' , IG is defined as the path integral of the gradients along the straight-line path from F to F' , which can be efficiently approximated numerically. In practice, IG for the i^{th} element of F is computed following numerical Riemman approximation

$$IG_i(F) ::= \frac{1}{m} (F_i - F'_i) \sum_{k=1}^m \frac{\partial IGNN(F' + \frac{k}{m}(F - F'))}{\partial F_i} \quad (11)$$

where F is the initial node feature matrix of graph data, F' is the starting point to calculate $IG_i(F)$ which is set to an all-zero matrix, $IGNN(\cdot)$ is the IGNN prediction, $\frac{\partial IGNN(\cdot)}{\partial F_i}$ is the gradient of IGNN prediction function relative to the i^{th} element of F , k is the scaled feature perturbation constant, and m is the number of steps in the Riemann approximation of the integral. With the patient-specific TACS graph data as input, IG reveals the relative prognostic importance of different TACS features at the ROI level."

06. Different histological subtypes of breast cancer present different collagen fibres architecture (doi: 10.1038/s41598-019-44156-9). Therefore, you should be careful with "[...] can be expanded to other reported in situ biomarkers, e.g. histological type[...]"

We have removed this sentence/claim in the revised manuscript as suggested.

In short, the great challenge for this manuscript is still to demonstrate reproducibility, even using artificial intelligence. However, despite the limitations, it is still an original manuscript and one that set out to develop something interesting in an extremely heterogeneous disease.

We have responded to Comment 03 on the reproducibility of cancer prognosis at individual and population levels. We sincerely thank the reviewer for overall positive assessment despite some limitations (which may be overcome in future more detailed studies).

REVIEWERS' COMMENTS

Reviewer #1 (Remarks to the Author):

The authors have responded to all of my comments. I do not have further queries.

Reviewer #2 (Remarks to the Author):

The authors have addressed my earlier concerns. Only one minor comment:

I would like to thank the authors for detailed clarification of the Bray-Curtis rank dissimilarities related to Figure S9. However, it's still not clear to me that why the ranks of Groupr are different in the same panel, given that all the three boxplots show the rank for sample pairs within Groups.

Reviewer #3 (Remarks to the Author):

The authors improve some critical points in the text. Although reproducibility is still a major challenge, the authors make it clear that this is a limitation of this study, and that it should be better addressed in further studies.

In my opinion, there are only one expression used in manuscript must be understood with caution.

1. lines 315-318: "The results demonstrated in this study promote a fundamental shift of disease prognosis toward underused in situ biomarkers, e.g. the histological type". The study focused on a single histological subtype - no special type (I guess), so I think this conclusion could lead a missinterpretation; even because the authors do not focus on the different histological subtypes (e.g. lobular, apocrine, tubular, mucinous, metaplastic...).

Although this study is innovative in the automated assessment of TACS/prognosis, it is still difficult to understand the impact that this will have due to the technology used and the lack of routine medical familiarity with those technology (MPM and/or machining learning) - as referred by reviewer #1.